# Centriolar satellites are acentriolar assemblies of centrosomal proteins

Valentina Quarantotti[1] , Jia-Xuan Chen[1,†] , Julia Tischer[1,†], Carmen Gonzalez Tejedo[1], Evaggelia K Papachristou[1], Clive S D'Santos[1], John V Kilmartin[2], Martin L Miller[1] & Fanni Gergely[1,*]

## Abstract

Centrioles are core structural elements of both centrosomes and cilia. Although cytoplasmic granules called centriolar satellites have been observed around these structures, lack of a comprehensive inventory of satellite proteins impedes our understanding of their ancestry. To address this, we performed mass spectrometry (MS)-based proteome profiling of centriolar satellites obtained by affinity purification of their key constituent, PCM1, from sucrose gradient fractions. We defined an interactome consisting of 223 proteins, which showed striking enrichment in centrosome components. The proteome also contained new structural and regulatory factors with roles in ciliogenesis. Quantitative MS on whole-cell and centriolar satellite proteomes of acentriolar cells was performed to reveal dependencies of satellite composition on intact centrosomes. Although most components remained associated with PCM1 in acentriolar cells, reduced cytoplasmic and satellite levels were observed for a subset of centrosomal proteins. These results demonstrate that centriolar satellites and centrosomes form independently but share a substantial fraction of their proteomes. Dynamic exchange of proteins between these organelles could facilitate their adaptation to changing cellular environments during development, stress response and tissue homeostasis.

**Keywords** centrioles; centrosome; composition; proteome; satellites
**Subject Categories** Cell Adhesion, Polarity & Cytoskeleton; Methods & Resources; Post-translational Modifications, Proteolysis & Proteomics
The EMBO Journal (2019) 38: e101082

See also: **L Gheiratmand et al** (July 2019)

## Introduction

In animal cells, the centrosome fulfils at least two distinct roles: in proliferating cells, it acts as the dominant microtubule-organising centre, whereas in non-dividing cells, it enables formation of the primary cilium, a sensory organelle that transduces chemical and mechanical signals (Spasic & Jacobs, 2017). Both these roles rely on tight control of centrosome numbers, achieved by a cell cycle-coupled centrosome assembly pathway (Banterle & Gonczy, 2017; Nigg & Holland, 2018). The centrosome comprises a centriole pair embedded within an ordered pericentriolar matrix (PCM). After division, each cell inherits two parental centrioles that differ in age and appearance with only the older (i.e. mother) centriole bearing appendages. During S phase, each centriole templates the growth of a single procentriole, a process dependent on polo-like kinase 4 (PLK4), CEP152, STIL, SAS-6 and CENPJ/CPAP (Banterle & Gonczy, 2017; Nigg & Holland, 2018). In G2, while procentrioles elongate, the daughter centriole matures by acquiring appendages, and by late mitosis, procentrioles disengage from their respective mothers and recruit PCM. Conversion of the mother centriole into a basal body underscores cilium assembly; distal and subdistal appendages facilitate recruitment of Golgi-derived vesicles onto the distal end of the mother centriole, which enable fusion with the plasma membrane and subsequent axoneme extension (Vertii et al, 2016). Consistent with the role of primary cilia in coordinating several signal transduction pathways including Hedgehog and PDGFR, cilia defects manifest as multisystemic genetic disorders and diseases (Spasic & Jacobs, 2017; Wang & Dynlacht, 2018).

Centriolar satellites are electron-dense, microtubule-associated, membraneless granules of ~ 70–100 nm that surround the centrosome and the basal body of the primary cilium (Bernhard & de Harven, 1960; de Thé, 1964; Sorokin, 1968; Steinman, 1968; Anderson & Brenner, 1971). Centriolar satellites have been observed both in proliferating and differentiated cells (Dammermann & Merdes, 2002; Vladar & Stearns, 2007; Espigat-Georger et al, 2016; Wang et al, 2016). The large coiled-coil protein Pericentriolar Material 1 (PCM1) was the first centriolar satellite constituent identified (Balczon et al, 1994; Kubo et al, 1999). Several additional centriolar satellite components have been found since either by co-localisation and/or co-immunoprecipitation with PCM1 (Tollenaere et al, 2015; Hori & Toda, 2017); some are exclusive to satellites (i.e. SSX2IP), whereas others are shared between satellites and centrosomes (i.e. CEP131, MIB1 and FOP), between satellites and cilia (i.e. BBS4) or between satellites, cilia and centrosomes (i.e. CEP290; Kim et al, 2004, 2008; Staples et al, 2012; Lee & Stearns, 2013; Villumsen et al, 2013).

1 Cancer Research UK Cambridge Institute, Li Ka Shing Centre, University of Cambridge, Cambridge, UK
2 MRC Laboratory of Molecular Biology, Cambridge Biomedical Campus, Cambridge, UK
*Corresponding author. Tel: +44 1223 769617; E-mail: Fanni.Gergely@cruk.cam.ac.uk
†These authors contributed equally to this work

PCM1 is an essential component of centriolar satellites, because its depletion or deletion causes many centriolar satellite proteins to lose their granular cytoplasmic appearance (Hori & Toda, 2017). PCM1 depletion disrupts radial organisation of microtubules and reduces centrosomal pools of ninein, centrin 3, pericentrin, NEK2A and CaMKIIβ (Dammermann & Merdes, 2002; Hames *et al*, 2005; Puram *et al*, 2011), while increasing centrosomal levels of CEP72, CEP90 and MIB1 (Young *et al*, 2000; Bärenz *et al*, 2011; Kim & Rhee, 2011; Stowe *et al*, 2012; Wang *et al*, 2016). In retinal pigment epithelial (RPE-1) cells, PCM1 is required for the assembly but not the maintenance of primary cilia (Kim *et al*, 2008; Sillibourne *et al*, 2013). PCM1 promotes cilia formation by preventing MIB1-driven degradation of the essential ciliary protein, TALPID3 (Wang *et al*, 2016). In addition, ciliogenesis is modulated by centriolar satellite components through their interaction with BBS4, a vital component of the ciliary membrane trafficking BBSome complex (Stowe *et al*, 2012; Zhang *et al*, 2012; Chamling *et al*, 2014).

Centriolar satellites are considered important regulators of centrosome and cilia function, but their precise contribution remains incompletely characterised. Identification of the centrosome proteome has transformed our understanding of centrosome function (Andersen *et al*, 2003; Jakobsen *et al*, 2011, 2013). To achieve a similar impact in centriolar satellite biology, we performed comprehensive proteomic profiling of satellites. Our study has identified 223 centriolar satellite-associated proteins of which 82 are known centrosomal proteins, although the proteome also contains several enzymes such as conserved E3 ubiquitin ligases. As demonstrated by quantitative methods, the majority of these proteins associate with centriolar satellites independent of centrosomes, although we identify a subset of centrosomal proteins with altered expression in acentriolar cells. Our study therefore reveals two distinct but co-existing subcellular pools of centrosomal proteins: centrosome- and centriolar satellite-associated ones. The latter could provide a reservoir of centrosomal proteins when transcription or translation of centrosomal genes is restricted.

# Results

### Centriolar satellite isolation from vertebrate lymphocytes

We set out to profile the proteome of centriolar satellites by mass spectrometry (MS), and define its dependency on centrosomes by comparing satellites of normal and acentriolar cells. DT40 chicken B lymphocytes were our cell line of choice, because we have already established two independent lines, which constitutively lacked centrioles due to deletions of STIL or CEP152 proteins (STIL-KO and CEP152-KO, respectively; Sir *et al*, 2013).

We based our centriolar satellite isolation method on a previous report that demonstrated successful enrichment of satellites from human cells using sucrose sedimentation followed by immunoprecipitation with anti-PCM1 antibodies (Kim *et al*, 2008). To avoid possible steric interference between PCM1 interactors and the PCM1 antibody, biallelic GFP tags were introduced into the C-terminus of PCM1 in wild-type (WT) and acentriolar DT40 cell lines (WT[PCM1-GFP], STIL-KO[PCM1-GFP] and CEP152-KO[PCM1-GFP] cell lines; Figs 1A and EV1A). On Western blots, PCM1-GFP levels were identical between the three cell lines (Fig 1B–D).

In WT cells, endogenous and GFP-tagged PCM1 appeared prominent around centrosomes (marked by γ-tubulin) with additional granules visible across the cytoplasm (Fig 1E). By contrast, only scattered granules were visible in ~ 50% of acentriolar cells, with the rest displaying a prominent PCM1 focus, which overlapped with γ-tubulin staining, and some additional granules (Fig 1F and G). As previously reported by our group, acentriolar cells contain transient γ-tubulin-positive assemblies that nucleate microtubules, and these could promote PCM1 and/or centriolar satellite clustering (Sir *et al*, 2013). We noted that co-treatment of cells with nocodazole and cytochalasin-B (Kim *et al*, 2008) improved the uniformity of PCM1 granule size between WT and acentriolar cells, and thus, we incorporated this step prior to cell lysis (Fig EV1B). This drug combination increased centriolar satellite clustering in acentriolar cells without causing satellite dispersal in WT. In sucrose density gradient centrifugation, PCM1-GFP and the centriolar satellite-associated ubiquitin ligase MIB1 were found in fractions 30–70% (Fig 1H). Centrosomes are expected to sediment in the higher sucrose fractions (Chavali & Gergely, 2015); indeed, the 60–70% fractions of WT cells appeared more enriched for the centriolar protein centrin 2 than the equivalent fractions in the acentriolar cells. To minimise co-isolation of centrosomes, only fractions 30–50% were pooled for subsequent GFP and control IgG immunoprecipitations (Fig 1H). On Western blots, pull-downs with GFP but not IgG antibodies yielded prominent bands of PCM1-GFP and MIB1. This method achieved highly specific co-immunoprecipitation of PCM1 protein complexes from pooled sucrose fractions, and it was therefore deemed suitable for mass spectrometry analysis.

### Centriolar satellite proteome revealed by label-free mass spectrometry-based quantification

Using the method in Fig 1H, centriolar satellites were isolated from WT[PCM1-GFP] and acentriolar STIL-KO[PCM1-GFP] and CEP152-KO[PCM1-GFP] cell lines (*n* = 6 for WT[PCM1-GFP]; *n* = 5 for STIL-KO[PCM1-GFP]; *n* = 4 for CEP152-KO[PCM1-GFP]; Fig 2A and B). Samples were processed for in-gel digestion followed by liquid chromatography-tandem mass spectrometry (LC-MS/MS) and analysed by label-free protein quantification using MaxQuant (Cox & Mann, 2008; Fig 2A). Briefly, following normalisation (Appendix Fig S1), proteins present in at least four of the six biological replicates from WT[PCM1-GFP], three out of the five replicates from STIL-KO[PCM1-GFP] or three out of the four replicates from CEP152-KO[PCM1-GFP] were retained, with the consideration that low abundant factors may not be detected in all MS runs. Specific PCM1-GFP interactors were determined based on a modified *t* statistic, taking both the intensity fold change and the paired *t*-test *P* value into account, for each protein in GFP and IgG pull-down. Label-free quantification revealed 223, 361 and 276 PCM1-associated proteins from WT[PCM1-GFP], STIL-KO[PCM1-GFP] and CEP152-KO[PCM1-GFP] cells, respectively (referred to as CS-WT, CS-STIL and CS-CEP152 proteomes hereafter where CS stands for centriolar satellite; Fig 2C, Table EV1). One hundred and seventy chicken proteins, and GFP, were shared between the three CS proteomes (Fig 2D). Results were validated by co-immunoprecipitation of several satellite candidates (e.g. CEP112, BICD2, WDR37) with PCM1-GFP from WT[PCM1-GFP] cells (Fig 2E).

Because CS-STIL and CS-CEP152 were derived from acentriolar cells, which could impact their overall composition, we used CS-WT

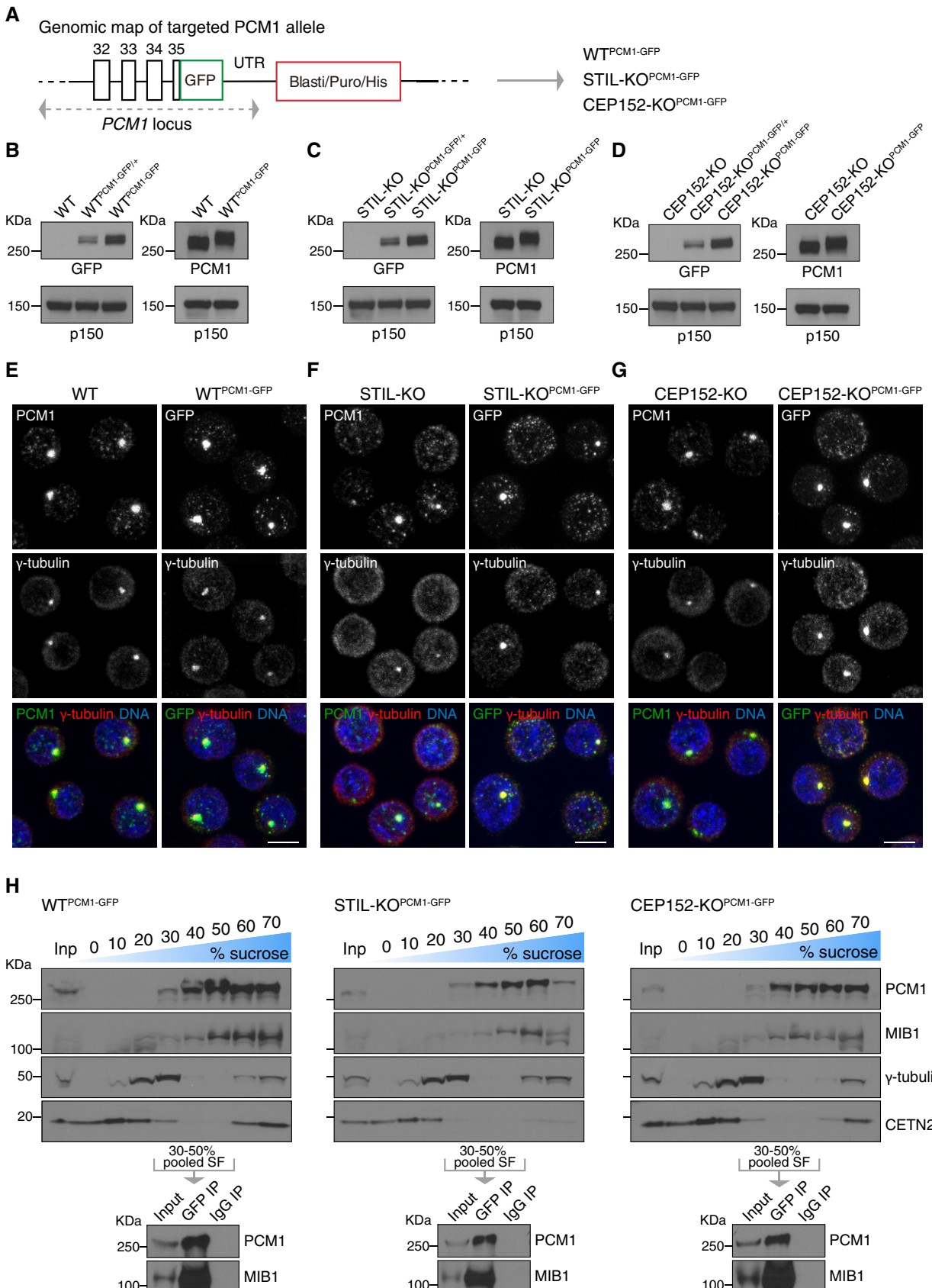

**Figure 1.**

◄

**Figure 1. Centriolar satellite isolation from wild-type and acentriolar DT40 cells.**

A Targeting of GFP into the *PCM1* locus, at the C-terminus. GFP was biallelically inserted in-frame with *PCM1* into WT, STIL-KO and CEP152-KO cells to obtain WT$^{PCM1-GFP}$, STIL-KO$^{PCM1-GFP}$ and CEP152-KO$^{PCM1-GFP}$ cells.

B–D Western blots of cytoplasmic extracts from WT (B), STIL-KO (C) and CEP152-KO (D) DT40 cells, probed with antibodies against GFP, PCM1 and the loading control, p150. Clones carrying mono- or biallelically GFP-tagged PCM1 alleles are denoted PCM1-GFP/+ and PCM1-GFP, respectively. Note the expected shift in PCM1 size in PCM1-GFP-targeted cells.

E–G PCM1-GFP phenocopies localisation pattern of untagged PCM1 in WT (E), STIL-KO (F) and CEP152-KO (G) cells. Representative immunofluorescence images of WT and WT$^{PCM1-GFP}$ cells, STIL-KO and STIL-KO$^{PCM1-GFP}$ and, CEP152-KO and CEP152-KO$^{PCM1-GFP}$ cells co-stained with antibodies against PCM1 (green) and γ-tubulin (red) or GFP (green) and γ-tubulin (red). DNA is in blue. Images correspond to maximum intensity projections of confocal micrographs. Scale bars: 5 μm.

H Upper panels depict Western blot analysis of PCM1, MIB1, γ-tubulin and centrin 2 (CETN2) sedimentation on 10–70% sucrose gradient of WT$^{PCM1-GFP}$ (left panel), STIL-KO$^{PCM1-GFP}$ (middle panel) and CEP152-KO$^{PCM1-GFP}$ (right panel) cells. 1% of the input and 5% of each sucrose fraction (SF) were loaded. 30–50% SF were pooled for immunoprecipitation with GFP antibody (GFP IP) or mouse IgG (IgG IP), and corresponding Western blots (lower panels) were probed with antibodies against PCM1 and MIB1.

as the reference proteome for all subsequent analyses. First, we intersected CS-WT with known centriolar satellite-associated proteins (Fig 3A). To date, a total of 51 proteins have been assigned to centriolar satellites by localisation (endogenous or overexpression) in various species and cell types (Gupta *et al*, 2015; Hori & Toda, 2017). Of these 51 proteins, 28 were found in CS-WT, a further 3 only in CS-STIL and/or CS-CEP152, whereas 3 were absent from the chicken genome (Fig 3B). Although present in both acentriolar satellite proteomes, the known satellite components CETN3 and CSPP1 were excluded from CS-WT because they did not pass the significance threshold (Fig 2C; Shearer *et al*, 2018). Second, we compared CS-WT with published interactomes of PCM1 obtained by immunoprecipitation (i.e. without prior sucrose gradient enrichment) or proximity-dependent biotinylation (BioID) using exogenously expressed FLAG-BirA*-PCM1 (Gupta *et al*, 2015). CS-WT contained 19 (of 48) proteins obtained by immunoprecipitation (PCM1-FLAG IP), and 43 (of 142) identified with BioID (PCM1-BioID; Fig EV2A). The sucrose sedimentation step seems important to attain larger PCM1-containing protein complexes, because FLAG-BirA*-PCM1 (PCM1-FLAG IP) co-immunoprecipitated with only 13 known centriolar satellite components (Fig EV2B and C). The total number of known components was similar between CS-WT and the protein network revealed by BioID, with 17 proteins found in both datasets. Thus, CS-WT shows substantial overlap not only with known centriolar satellite proteins but also with PCM1 interactors obtained by proximity labelling.

There are also notable differences between the various datasets and the known centriolar satellite component list; these may be due to differences in species, cell types and methodologies. Moreover, satellite granules are highly heterogeneous in size even within the same cell, a feature evident both by microscopy and sucrose sedimentation (Fig 1E–G; Kim *et al*, 2008). CS-WT proteome therefore corresponds to the collective content of PCM1-containing granules whose individual composition may differ.

### Marked enrichment of centrosomal proteins in CS proteome

We next performed GO enrichment analysis on the CS-WT proteome; centrosome, centriolar satellites and microtubules were the most over-represented cellular compartments, whereas ciliary basal body docking and centrosome organisation the most over-represented biological processes (Fig 3C). Indeed, when comparing our dataset to published proteomic data of human centrosomes (Jakobsen *et al*, 2011), 82 of the 223 proteins corresponded to

conserved centrosomal proteins, most of which without previous evidence of centriolar satellite association (Fig 3D). Furthermore, the overlap between CS-WT and the interactome obtained by PCM1 proximity labelling (PCM1-BioID) consisted predominantly of centrosomal proteins (Fig EV2D; Gupta *et al*, 2015). Components of all substructures of centrosomes were represented in CS-WT (e.g. PCM, G1 linker, appendages), and the dataset also included proteins with roles in centriole assembly and elongation (Fig 3E).

Importantly, the 170 proteins shared between CS-WT, CS-STIL and CS-CEP152 still included 28 known satellite components and 79 centrosomal proteins, indicating that the presence of centrosomal proteins in CS-WT is not due to co-precipitation of intact centrosomes with PCM1-GFP. GO enrichment analysis of the 61 proteins shared only between CS-STIL and CS-CEP152 (absent from CS-WT) failed to reveal any specific pathway. Intriguingly, however, this set contained several kinases with centrosome-related functions such as CSNK1D, ILK, PLK1 and STK3 (Table EV1).

In addition, in CS-WT we identified several new centriolar satellite-associated regulatory factors, microtubule motors and adaptors, and consistent with a role for satellites in ciliogenesis, also ciliopathy-linked proteins and modulators of Hedgehog signalling (Figs 3E and EV2E; Breslow *et al*, 2018; Lee *et al*, 2012; Sergouniotis *et al*, 2014).

### Centriolar satellite candidate proteins co-immunoprecipitate and co-localise with PCM1 in human cells

The CS-WT proteome contained orthologues of 28 proteins that were previously identified as centriolar satellite components in mammalian cells, indicative of evolutionary conservation between chicken and human satellite composition. We further tested PCM1 association of satellite candidates selected from the 170 proteins shared between CS-WT, CS-STIL and CS-CEP152 (Fig 2D and Table EV1). PCM1-binding complexes were isolated from human embryonic kidney (HEK293T) cells using sucrose sedimentation followed by immunoprecipitation with anti-PCM1 antibodies (Fig 4A). Western blots were probed with antibodies against candidate proteins, including known centrosome components (CEP112, CEP170), a microtubule motor adaptor (BICD2), E3 ubiquitin ligases (HERC2, MYCBP2) and T3JAM, a protein implicated in autophagy and T-cell development, along with positive controls (PCM1, CEP290 and DZIP1; Kim *et al*, 2008; Zhang *et al*, 2017). All candidates co-immunoprecipitated with endogenous PCM1 from HEK293T cells (Fig 4A).

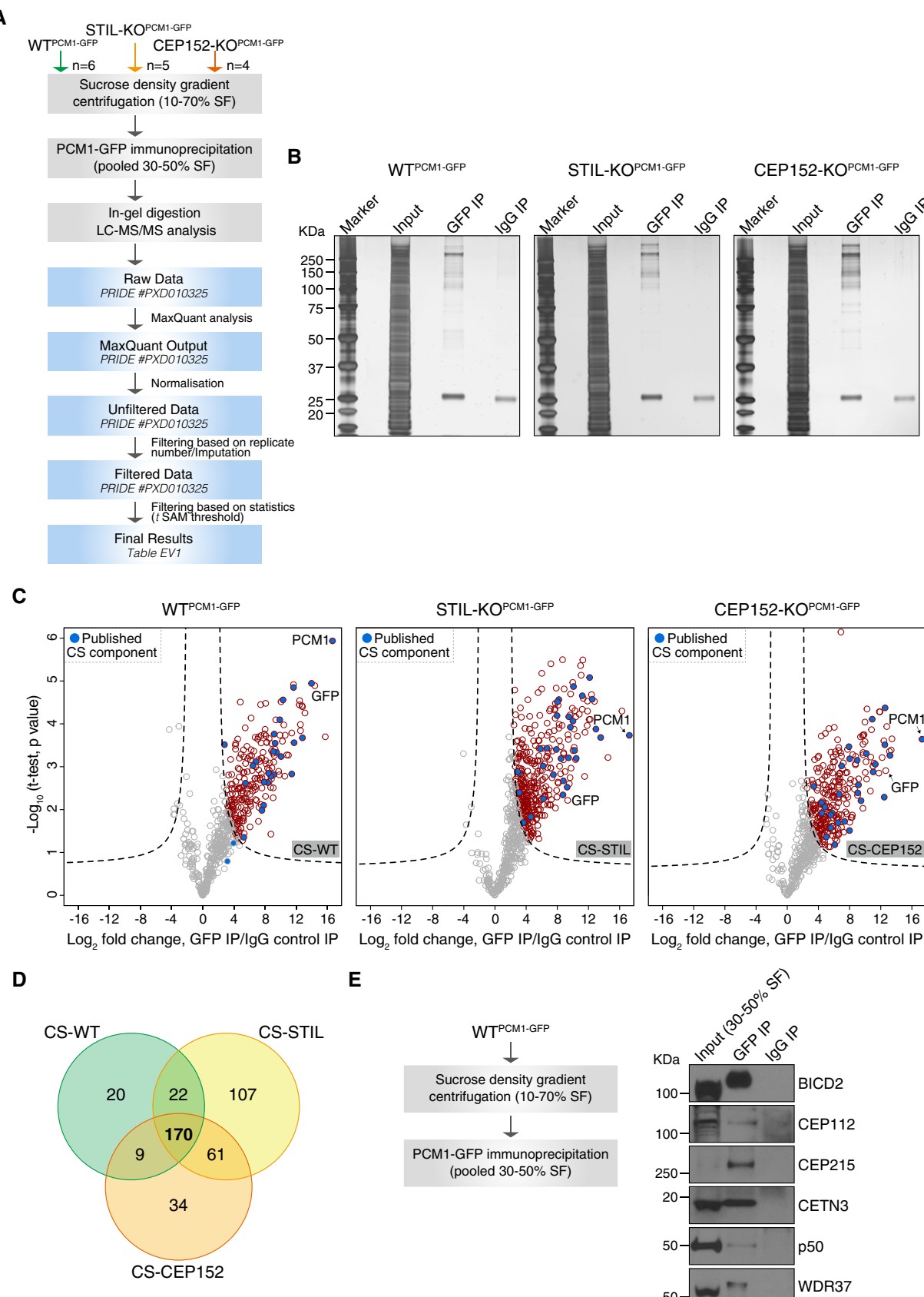

**Figure 2.**

◄

**Figure 2.  Label-free mass spectrometry analysis of centriolar satellite proteomes from wild-type and acentriolar cells.**

A   Experimental workflow of satellite isolation and proteomic analysis. Number of biological replicates per genotype is indicated. SF: sucrose fraction.

B   Representative silver-stained gels of GFP or IgG immunoprecipitation (IP) from pooled 30–50% sucrose fractions of WT[PCM1-GFP], STIL-KO[PCM1-GFP] and CEP152-KO[PCM1-GFP] cells. 0.5% of input and 5% of the IP samples were loaded.

C   Volcano plots showing quantitative label-free mass spectrometry data from WT[PCM1-GFP] (left panel), STIL-KO[PCM1-GFP] (middle panel) and CEP152-KO[PCM1-GFP] (right panel) cell lines. Red circles correspond to hits significantly enriched in GFP vs. IgG pull-downs. Previously described satellite components, including PCM1, are shown in blue. Data were filtered to retain proteins detected in a minimum of 4/6 (CS-WT), 3/5 (CS-STIL) and 3/4 (CS-CEP152) replicates.

D   Venn diagram of the number of candidates identified in CS-WT, CS-STIL and CS-CEP152. Multiple gene products associated with the same gene symbol have been collapsed into one.

E   PCM1 co-immunoprecipitates with satellite candidates in DT40 cells. Workflow is shown on the left. Western blots on right depict immunoprecipitation performed using a GFP antibody (GFP IP) or mouse IgG (IgG IP) on pooled 30–50% sucrose fractions from WT[PCM1-GFP] cells. Blots were probed with antibodies as indicated. 1% of the input and 5% of the pull-down samples were loaded.

We next tested localisation of known (i.e. SSX2IP, CEP63) and putative new components of centriolar satellites in human Jurkat T lymphocytes (Fig EV3A). Like DT40 cells, Jurkat cells exhibit a predominantly pericentrosomal PCM1 signal, which diminishes upon microtubule depolymerisation (Fig EV3B). The E3 ubiquitin ligases (HERC2, MYCBP2 and TRIM41) did not show any specific localisation pattern in Jurkat cells, but we confirmed near-complete co-localisation of PCM1 with T3JAM and SPICE1 (Figs 4B and EV3A). As expected, centrin 2 staining appeared similar to that of PCM1 (Dammermann & Merdes, 2002), whereas a lesser degree of overlap was seen with BICD2, CEP170, CDK5RAP2/CEP215 and WDR90 with all four showing prominent centrosomal signal. CP110 and CEP63 appeared exclusively on centrioles. In RPE-1 cells, only T3JAM showed extensive co-localisation with PCM1, and this signal diminished upon depletion of T3JAM by siRNA (Fig 4B and C). In RPE-1 cells, satellite levels of the other candidates may be below detection limits as it was previously seen for CEP63 (Firat-Karalar *et al*, 2014), whereas in lymphocytes the greater concentration of PCM1 in the pericentrosomal region could facilitate detection. Indeed, when exogenously expressed, GFP-TRIM37, GFP-CEP170 and GFP-CCDC77 co-localised with PCM1 in 293 cells (Fig 4D and Appendix Fig S2). In summary, we confirmed co-immunoprecipitation and/or co-localisation between PCM1 and several new centriolar satellite candidates in multiple human cell lines (for overview see Fig 5D).

## Centriolar satellite-associated ubiquitin ligases promote ciliogenesis

CS-WT contains both suppressors (i.e. MIB1) and activators of ciliogenesis (i.e. CEP290, CBY1, TALPID3), and essential ciliary transport proteins (i.e. BBSome, IFT74;Wang & Dynlacht, 2018). In addition to MIB1, the E3 ubiquitin ligases CUL3, HERC2, MYCBP2, TRIM37 and TRIM41 are also enriched in CS-WT. TRIM37 has been linked to centrosome duplication and survival of p53-competent acentriolar cells (Balestra *et al*, 2013; Fong *et al*, 2016; Lambrus *et al*, 2016; Meitinger *et al*, 2016), whereas HERC2 mediates centrosome integrity (Al-Hakim *et al*, 2012). However, except for CUL3, which was found to promote cilia formation, the function of the other ubiquitin ligases in ciliogenesis has not yet been established (Kasahara *et al*, 2014).

We therefore depleted these ubiquitin ligases by RNA interference and enumerated cilia in serum-starved RPE-1 cells (Fig 5A). Cilia formation was consistently reduced by ~ 40% in HERC2- and MYCBP2-depleted cells, with TRIM41 depletion causing a milder

defect. The impact of TRIM37 depletion on ciliogenesis is inconclusive, since there was no consensus between the three siRNAs, despite all effectively depleting *TRIM37* mRNA (Fig EV3C). In addition to ubiquitin ligases, we evaluated the role of structural centriolar satellite candidates in cilia assembly. Of these, depletion of BICD2 and CCDC77 markedly suppressed cilia formation in RPE-1 cells (Fig 5A). A reduction in cilia numbers was noted in cells treated with CEP170 siRNA 2 (si2) but not siRNA1 (si1); since both effectively depleted CEP170, we cannot exclude off-target effects (Figs 5A and EV3C). Reduced ciliogenesis was also seen in cells treated with T3JAM siRNA 2 (si2), the siRNA that effectively depleted T3JAM (Figs 5A and EV3C). Consistent with a function in ciliogenesis, T3JAM was detectable in satellites surrounding the ciliary base (Fig 5B). To assess whether the decrease in cilia numbers was due to impaired cilia growth, we examined cilia length upon depletion of satellite candidates (Fig 5C). Shorter cilia were observed in cells depleted of BICD2, HERC2 and TRIM41 with a milder effect in MYCBP2-depleted cells.

We therefore conclude that of the PCM1-associated E3 ubiquitin ligases, TRIM41 promotes cilia elongation, whereas HERC2 and MYCBP2 facilitate formation and elongation of cilia (Fig 5D).

## A subset of conserved centrosomal proteins is down-regulated in acentriolar cells

Acentriolar chicken cells show a delay in mitosis and increased chromosomal instability (Sir *et al*, 2013), but how centriole loss affects the cellular proteome, and in particular, the levels of centrosomal proteins, has not yet been investigated. Given that nearly half of the known centrosome proteome can also be found in centriolar satellites (Fig 3D), one possibility is that in acentriolar cells surplus centrosomal proteins become incorporated into centriolar satellites, thereby increasing their representation within the satellite proteome. Conversely, centrosomes may promote satellite recruitment of centrosomal proteins, which would result in satellite association of these factors in acentriolar cells. To determine the effect of centriole loss on satellite association of individual proteins, we first had to assay total cellular level of proteins by performing quantitative proteomic profiling on whole-cell proteomes of wild-type and acentriolar cells (Figs 6 and EV4).

Using stable isotope labelling by amino acids in cell culture (SILAC)-based proteomics, we compared whole-cell proteomes (WCP) of differentially labelled WT[PCM1-GFP] and STIL-KO[PCM1-GFP] cells (*n* = 6 replicates; four forward and two reverse SILAC label-swap experiments; Figs 6A and EV4A). In total, we identified 7,070

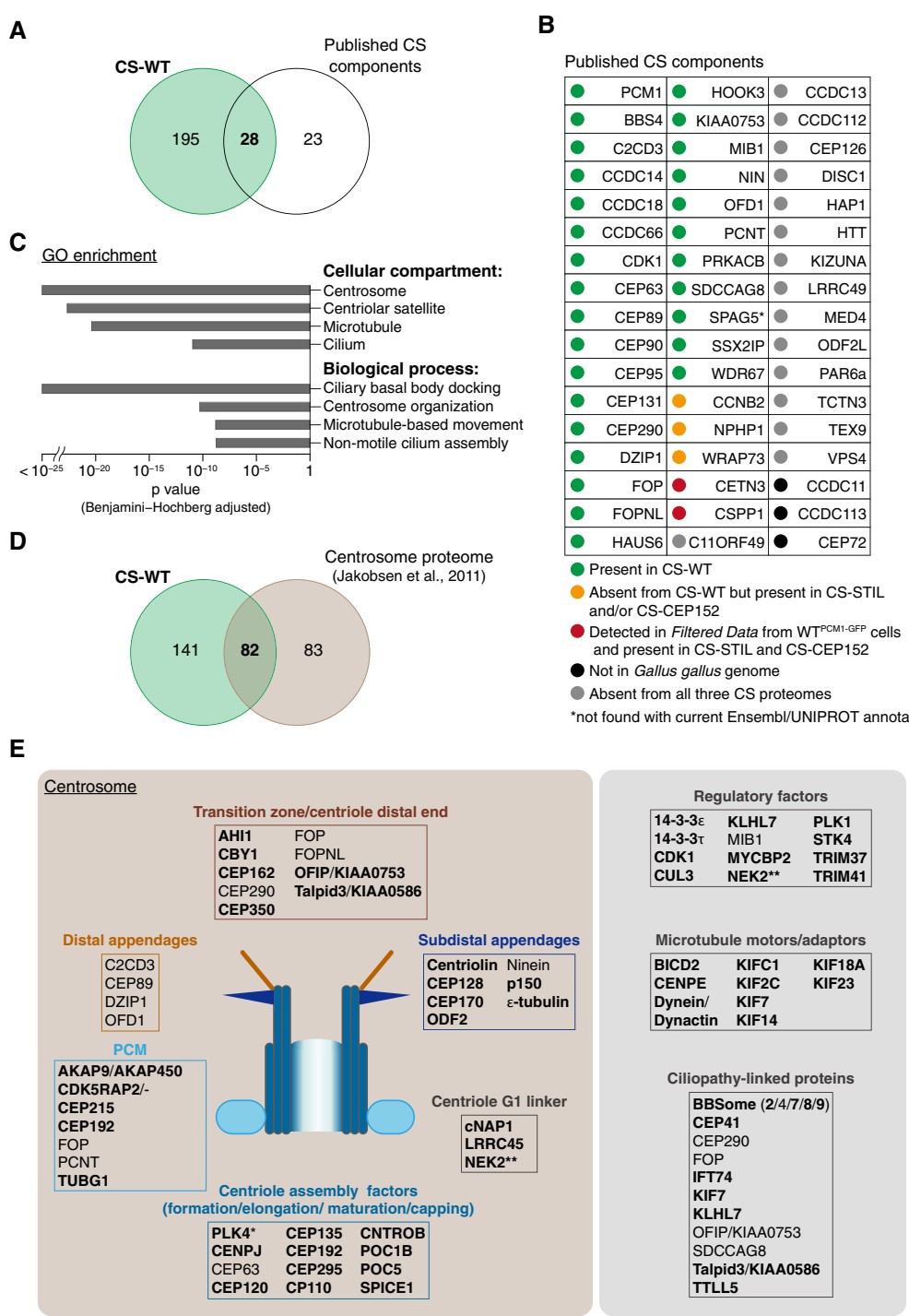

**Figure 3. The centriolar satellite proteome is enriched in centrosome- and cilia-related proteins.**

A Venn diagram of total protein numbers from CS-WT and known satellite components. Note that this and all subsequent analyses were performed on human orthologues of the chicken proteins from CS-WT.

B Table depicts previously reported satellite components and their status in CS-WT. Note that the current Ensembl/UNIPROT annotation fails to detect SPAG5.

C Gene Ontology (GO) enrichment analysis performed on CS-WT. Selected terms with high significance are shown with the corresponding Benjamini–Hochberg adjusted P values.

D Venn diagram of total protein numbers from CS-WT and the comprehensive proteomic dataset of centrosomes as reported in Jakobsen et al (2011).

E Schematics depict a selection of known and newly identified (bold) satellite candidates (from CS-WT) and the centrosomal substructures and/or pathways they are implicated in. PLK4 is marked with an asterisk, as it was detected only in CS-STIL. Several ciliopathy-associated proteins are found in CS-WT including newly identified candidates (bold). NEK2 is marked with two asterisks, as it was also shown to partially co-localise with PCM1 by Hames et al (2005).

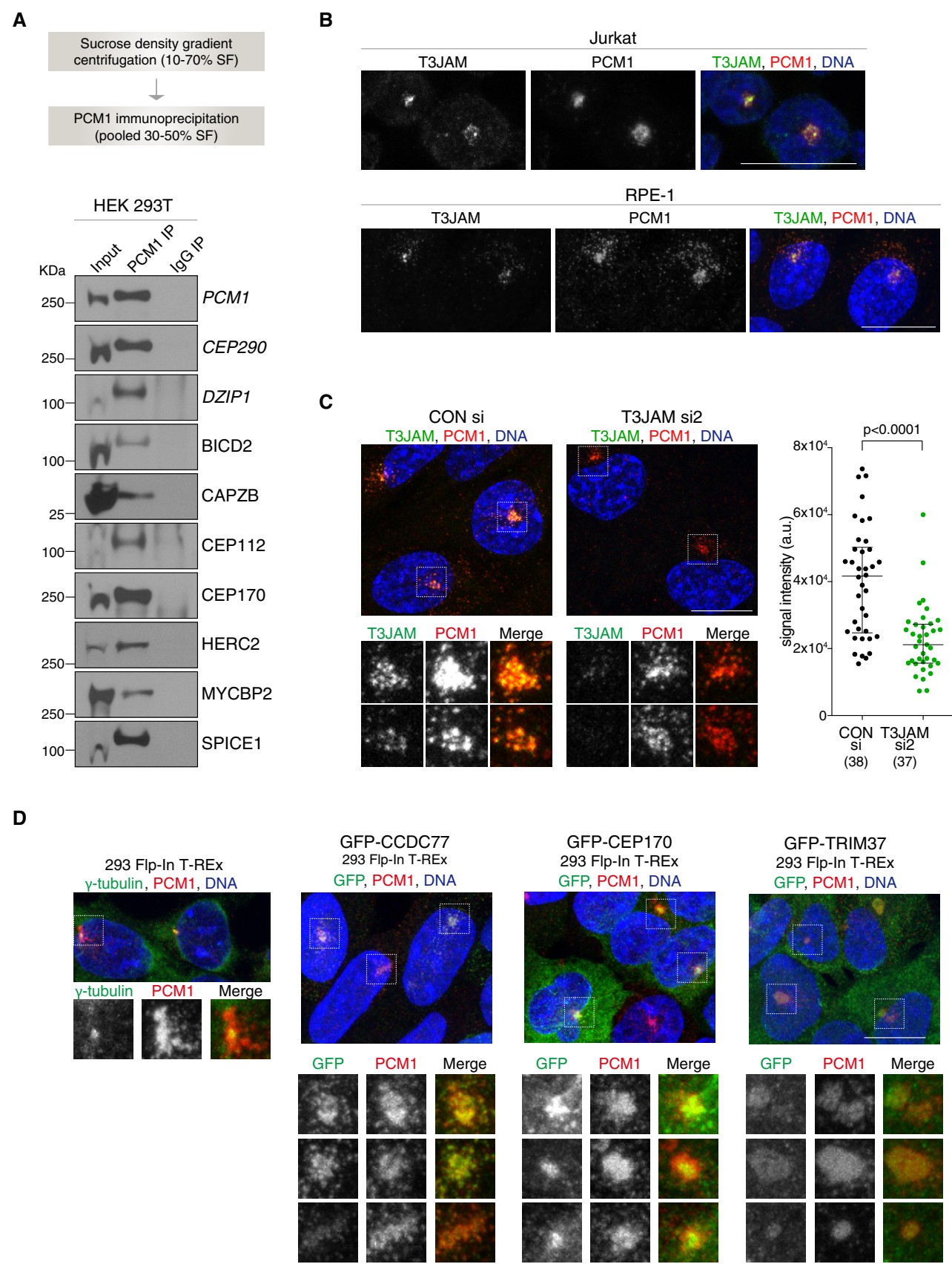

**Figure 4.**

**Figure 4. Selected centriolar satellite candidate proteins associate and co-localise with PCM-1 in human cells.**

A  Workflow of satellite isolation from HEK293T cells is shown on top. Western blots below depict immunoprecipitation performed using a PCM1 antibody (PCM1 IP) or mouse IgG (IgG IP) on pooled 30–50% sucrose fractions (input). Blots were probed with antibodies against satellite candidates with CEP290 and DZIP1 serving as positive controls (shown in italic). 1% of the input and 5% of the pull-down samples were loaded.

B  Representative immunofluorescence images of T3JAM localisation in Jurkat (top panels) and RPE-1 (bottom panels) cells. Cells were co-stained with antibodies against PCM1 (red) and T3JAM (green). DNA is in blue. Images correspond to maximum intensity projections of confocal micrographs. Scale bar: 10 μm. Immunofluorescence images of further candidates are shown in Fig EV3A.

C  Representative immunofluorescence images of RPE-1 cells treated with control (CONsi) or T3JAM-targeting (T3JAMsi2) siRNAs. Satellites are shown at high magnification below. Cells were co-stained with antibodies against PCM1 (red) and T3JAM (green). DNA is in blue. Images correspond to maximum intensity projections of confocal micrographs. Scale bars: 10 μm. Graph on right depicts T3JAM total signal intensities measured across a circle ($d$ = 8 μm) encompassing PCM1 signal. Medians and interquartile ranges are indicated; the number of cells analysed is reported in parentheses. $P$ values were obtained by Mann–Whitney $U$ test.

D  Representative immunofluorescence images of parental 293 Flp-In T-REx cells and cells expressing GFP fusions of satellite candidates. Satellites are shown at high magnification below. Cells were co-stained with antibodies against GFP (green) and PCM1 (red) following treatment with 4 μg/ml tetracycline (see Western blots in Appendix Fig S2). DNA is in blue. Images correspond to the maximum intensity projection of the confocal micrograph. Note smooth appearance of PCM1 signal upon TRIM37 overexpression. Scale bars: 10 μm.

proteins and quantified SILAC ratios for 6,197 proteins with a false discovery rate of 1% (Table EV2). One hundred and twelve (of the 165 known) centrosomal proteins were quantified despite their overall low abundance. Interestingly, the 25 most down-regulated proteins in STIL-KO$^{PCM1-GFP}$ cells included five centrosomal proteins in addition to STIL (Fig 6B and C). Levels of only two centrosomal components were elevated in STIL-KO$^{PCM1-GFP}$ cells: PLK4 and TRAF-5 (Fig 6B and D). The rise in PLK4 is consistent with reports of increased cytoplasmic and centrosomal PLK4 levels in STIL-deficient cells (Arquint *et al*, 2015; Moyer *et al*, 2015).

Our data revealed changes in a number of proteins without obvious links to centrosome biology; as shown in Fig 6D, an increase of over 5-fold was seen in levels of MAP kinase-interacting serine/threonine kinase 1 (MKNK1), whereas ribosomal protein RPL22L1 was reduced 10-fold (Fig 6C). Since the WCP was obtained from a single clone of acentriolar STIL-KO cells, we cannot exclude that some of these differences are due to clonal expansion or lack of STIL (e.g. PLK4) rather than centriole loss. Nonetheless, given the reduction in the levels of at least five centrosomal proteins, none of which are known binding partners of STIL (e.g. CEP57 or CEP63), the majority of changes are likely to be attributable to the acentriolar state.

## The centriolar satellite proteome is largely unaltered in acentriolar cells

To determine whether satellite association of proteins is altered in acentriolar cells, we performed SILAC analysis of CS derived from WT$^{PCM1-GFP}$ and STIL-KO$^{PCM1-GFP}$ cells, following the experimental workflow in Fig 6A (SILAC-CS; $n$ = 3 replicates: two forward and one reverse SILAC label-swap experiments; Figs 6E, and EV4B and C). Altogether, we identified 870 proteins and quantified SILAC ratios for 502 proteins with a false discovery rate of 1% (Table EV3). One hundred and seventy-seven proteins from the original 223 in CS-WT were quantified in SILAC-CS. Furthermore, levels of PCM1 were comparable between normal and acentriolar cells both in WCP (log$_2$ mean ratio STIL/WT = −0.02; Significance B = 0.98) and CS (log$_2$ mean ratio STIL/WT = −0.07; Significance A = 0.84), thus confirming the validity of the approach.

Twenty-five proteins were enriched, and 47 proteins (including 29 centrosomal factors) were reduced in SILAC-CS-STIL (Fig EV4D and E). For most proteins, the trend reflected results from the WCP

(i.e. PLK4 is up-, whereas CEP63 and CEP57 down-regulated) with the ciliopathy-associated proteins CEP41 and CENPF being notable exceptions (Lee *et al*, 2012; Waters *et al*, 2015). Despite no change in overall protein levels, CEP41 was 16-fold down-regulated in SILAC-CS-STIL, whereas CENPF exhibited a 2-fold increase in SILAC-CS-STIL (Fig 6E–G). In line with these findings, CENPF was detected in both CS-STIL and CS-CEP152, but not in CS-WT, whereas CEP41 was present in CS-WT but absent from satellites of acentriolar cells (Table EV1). CEP41 has been shown to bind and regulate entry of a tubulin polyglutamylase enzyme, TTLL6, a step critical for tubulin glutamylation of the ciliary axoneme (Lee *et al*, 2012). Interestingly, CS-WT contains another family member TTLL5, and thus, CEP41 may associate with TTLLs on satellites.

Collectively, these results indicate that centriolar satellites form independently of centrosomes and harbour a distinct pool of centrosomal proteins.

## The role of PCM1 in steady-state expression and localisation of satellite proteins

So far, we have addressed the consequences of centriole loss on whole-cell and centriolar satellite proteomes. However, given the substantial overlap between satellite and centrosome proteomes, we also wanted to test whether centriolar satellites could regulate cellular and/or centrosomal levels of satellite candidates.

To this end, multiple PCM1 knock-out RPE-1 cell clones were generated using CRISPR/Cas9 (called KO1-4; Fig EV5A and B). Consistent with previous reports (Wang *et al*, 2016), KO clones exhibited a marked ciliogenesis defect (Fig EV5C). We first examined steady-state protein expression levels of several known centriolar satellite components and new candidates in the KO clones. In line with previous studies, only two known components were affected by PCM1 loss: a marked reduction was seen in levels of SSX2IP, whereas MIB1 levels were elevated (Fig 7A and Appendix Fig S3; Klinger *et al*, 2014; Wang *et al*, 2016). Because steady-state levels of satellite candidates were largely unaffected in PCM1-KO cells, we next assayed if PCM1 loss influenced their subcellular distribution. Centriolar satellite localisation of T3JAM was lost in PCM1-KO cells, with T3JAM instead adopting a weak but tight centriolar localisation (Fig 7B). For those centriolar satellite candidates that associate with centrosomes in RPE-1 cells, centrosomal levels were quantified by measuring signal intensities

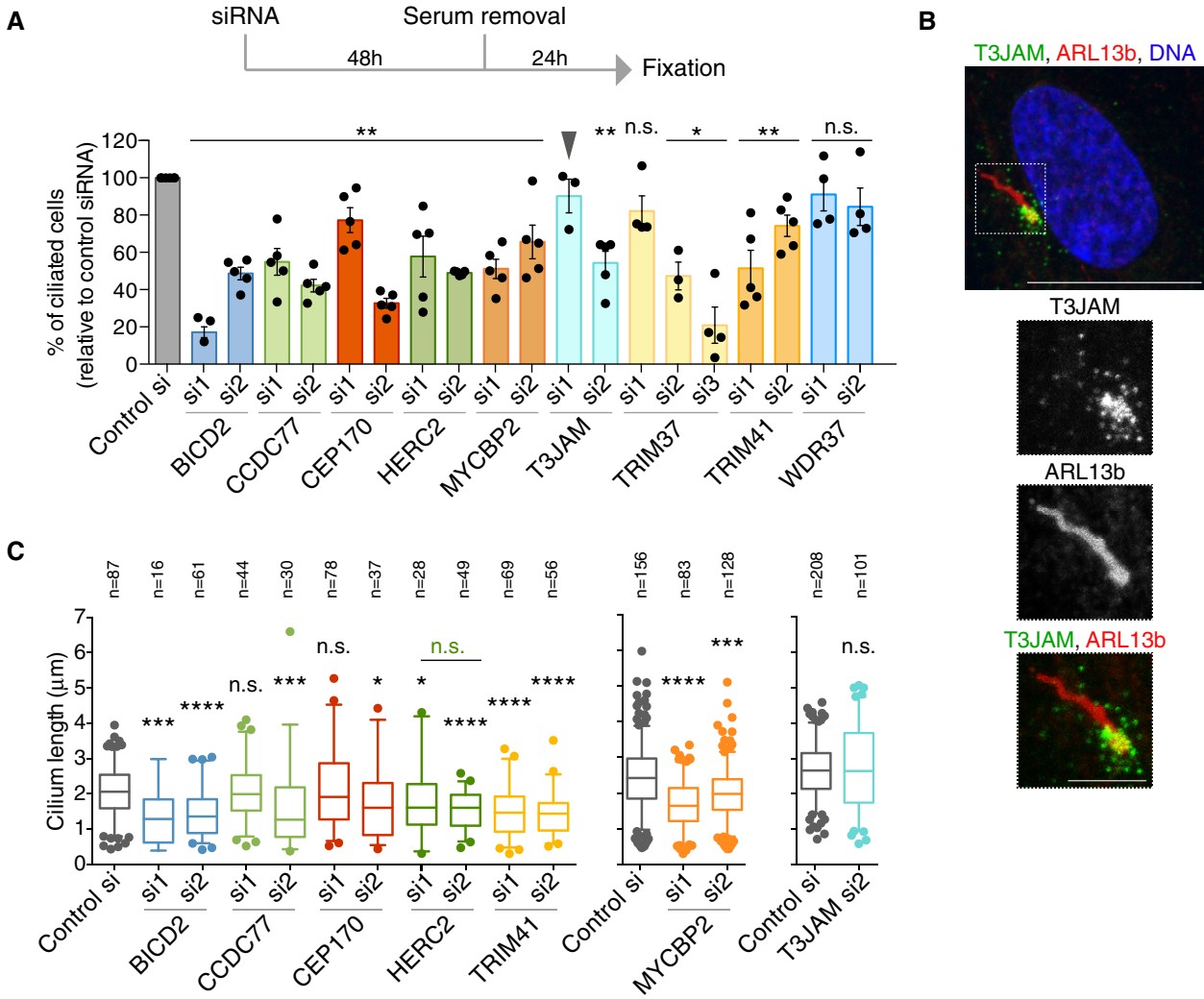

Figure 5.

**Figure 5.** **Selected centriolar satellite candidate proteins positively regulate ciliogenesis.**

A   Evaluating the roles of satellite candidates in ciliogenesis by siRNA-mediated depletion. Experimental timeline is depicted on top with graph below showing the percentage of ciliated cells relative to control siRNA (si) treatment. Datapoints correspond to biological replicates (n = 3–5 per siRNA, 100–200 cells scored per experiment; for HERC2 si, 60–100 due to reduced viability). P values were obtained by Mann–Whitney U test and are relative to control siRNA (*P ≤ 0.05; **P ≤ 0.01; n.s.: not significant). Note that T3JAMsi1 failed to deplete *T3JAM* (arrowhead) according to Fig EV3C. Bar chart depicts mean percentage ± SE.

B   Representative immunofluorescence image shows T3JAM localisation at the base of the primary cilium in serum-starved RPE-1 cells. Cells were co-stained with antibodies against T3JAM (green) and the ARL13b (red). DNA is in blue. The image corresponds to maximum intensity projection of the confocal micrograph. Scale bar: 10 μm. Scale bar on cilia zoom in: 3 μm.

C   Cilia length was determined in cells depleted of candidate satellite components. Datapoints were collected from two biological replicates; total number of cilia scored is indicated. In the plots, boxes represent interquartile ranges, horizontal lines the medians, and whiskers the 5th–95th percentiles. P values were obtained by Mann–Whitney U test (*P ≤ 0.05; ***P ≤ 0.001; ****P ≤ 0.0001).

D   Table provides an overview of validation and functional assays performed on satellite candidates. IF: immunofluorescence. For proteins where antibody epitope showed high conservation between human and chicken, the antibody was tested on DT40 cells. Inconclusive (inc.) refers to non-specific bands on Western blots or diffuse staining in IF.

within a fixed-size circle encompassing the centrosome (Fig 7C). As expected from previous work, centrosomal pools of centrin 3, ninein and pericentrin were reduced in PCM1-KO cells (Dammermann & Merdes, 2002); however, levels of centrosomal CEP215, CEP128 and CP110 were also lower (Fig 7C, see Fig 7D for summary). This could reflect a genuine decrease in centrosomal targeting of these proteins, but it is also feasible that pericentrosomal satellite clusters overlap with the centrosomal region, thereby hindering our analysis. Even if the latter is the case, the results indicate that a PCM1-dependent pool of these factors exists within close proximity of centrosomes. We also noted a previously unreported decrease in γ-tubulin levels in PCM1-KO clones, especially during interphase (Fig 7C), which may be a secondary consequence of reduced centrosomal levels of pericentrin involved in centrosomal recruitment of γ-tubulin (Lin *et al*, 2015; Gavilan *et al*, 2018).

In summary, the concomitant increase in MIB1 (suppressor of cilia formation) and reduction in SSX2IP (activator of ciliogenesis) could collude to block ciliogenesis in PCM1-KO RPE-1 cells, but effects of PCM1 loss on proliferation (Fig EV5D), microtubule organisation and centrosomal targeting of proteins may also contribute to the overall phenotype. Because centriolar satellites are likely to contain cell type- and tissue-specific components, it will be important to establish whether PCM1 plays distinct roles in the physiology of different cell types and tissues.

# Discussion

In this study, we have identified 223 putative centriolar satellite components from chicken lymphocytes (CS-WT). Centriolar satellite association has already been reported for 29 of these 223 factors, and we have validated a further 10 by immunoprecipitation and/or immunofluorescence in human cell lines. Importantly, 170 proteins were shared between CS-WT and centriolar satellites from acentriolar cell lines, implying that cells contain highly reproducible PCM1-associated protein assemblies, which form independently of centrioles.

Quantitative SILAC analysis uncovered relatively few differences between satellite composition of WT and STIL-KO cells; however, two aspects of the experimental design could have caused compression of SILAC ratios. First, satellite isolation was performed from cells with partially depolymerised microtubule and actin networks, which may have masked some of the differences between control and acentriolar satellite proteomes. Second, the mixing of light- and heavy-labelled lysates prior to satellite purification potentially enabled exchange between differentially labelled forms of satellite components, thus compressing SILAC ratios of dynamic interactors of PCM1 (Mousson *et al*, 2008). Nonetheless, for the subset of proteins we could quantitate both in whole-cell proteomes and centriolar satellites of WT and STIL-KO cells, the changes in total levels did not exceed those in satellite pools. Interestingly, despite

**Figure 6.** **SILAC-based quantitative analysis of whole-cell and satellite proteomes of acentriolar cells.**

A   Experimental workflow of SILAC-based MS analysis. Note that differentially labelled cells were mixed prior to the multistep CS purification process to minimise variability in sample preparation.

B   Comparison of whole-cell proteomes (WCP) of differentially labelled WT^PCM1-GFP and STIL-KO^PCM1-GFP cells. Scatterplot shows the fold change in SILAC protein ratios. Proteins with significant changes in abundance between WT^PCM1-GFP and STIL-KO^PCM1-GFP cells are shown as red circles [determined using Significance B with protein ratios stratified according to protein intensity (Cox & Mann, 2008)], whereas proteins previously identified in CS-WT are in blue.

C   Table shows list of proteins reduced in WCP-STIL, ranked according to fold change. In addition to the two proteins with the greatest log₂ fold change, all significantly down-regulated centrosomal and/or satellite proteins are included. Note that the "re-quantify" function of MaxQuant enabled assignment of SILAC ratios to proteins with infinity ratios (i.e. STIL in STIL-KO).

D   Table shows list of proteins elevated in WCP-STIL, ranked according to fold change. PLK4 is the most significantly up-regulated centrosomal protein.

E   Comparison of satellite proteomes isolated from differentially labelled WT^PCM1-GFP and STIL-KO^PCM1-GFP cells. Scatterplot shows the fold change in SILAC protein ratios. The x-axis corresponds to the forward, whereas the y-axis to the reverse experiments. Proteins with significant changes in abundance between satellite isolated from WT^PCM1-GFP and STIL-KO^PCM1-GFP cells are shown as red circles [obtained by Significance A parameter (Cox & Mann, 2008)], whereas those previously identified in CS-WT are in blue.

F   Table shows list of proteins reduced in SILAC-CS-STIL and their corresponding fold change in the WCP. They are ranked according to fold change. Changes that are not significant are depicted in italics; n.d.: not detected.

G   Table shows list of proteins elevated in SILAC-CS-STIL and their corresponding fold change in the WCP. They are ranked according to fold change. Changes that are not significant are depicted in italics.

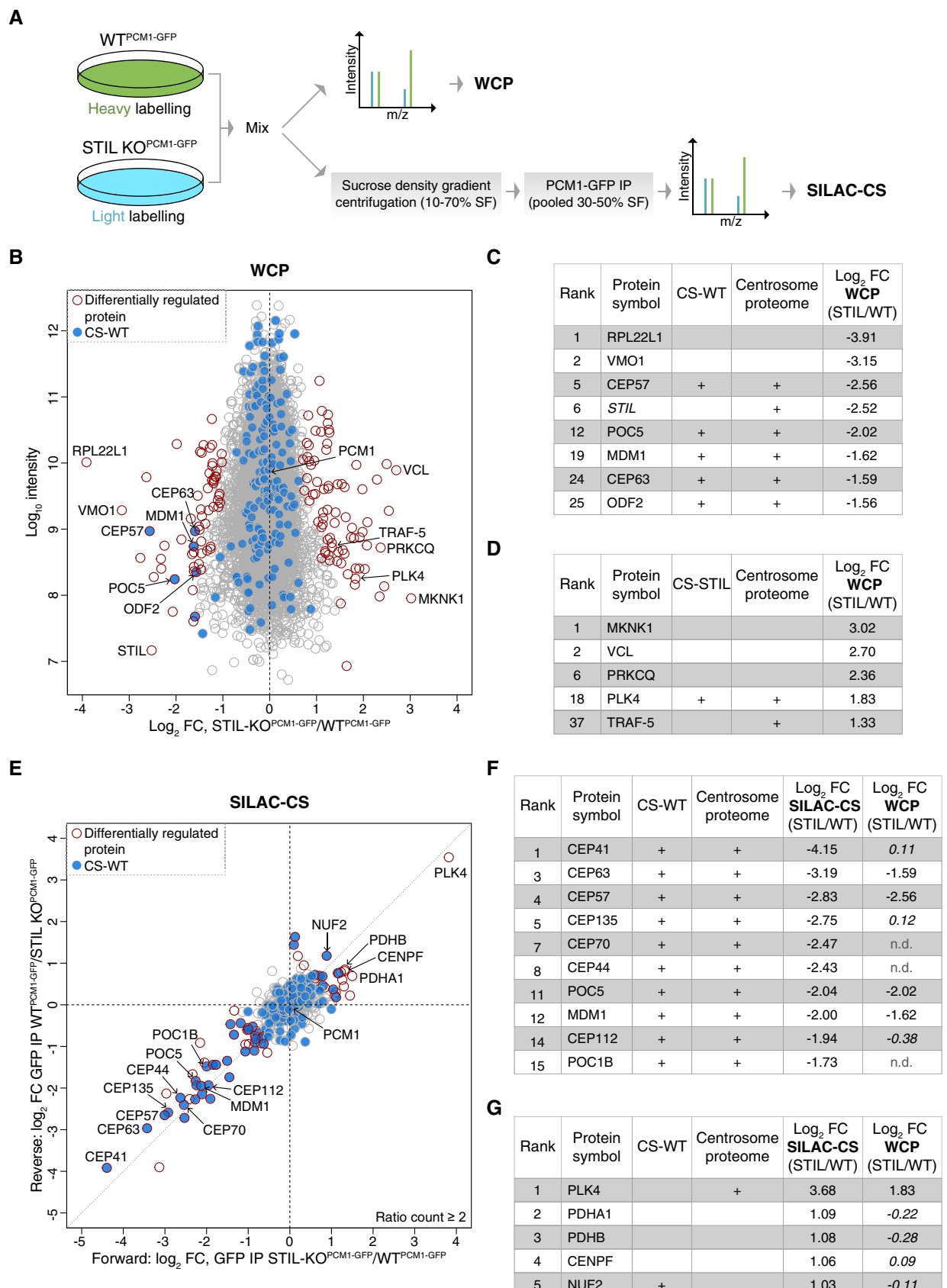

**Figure 6.**

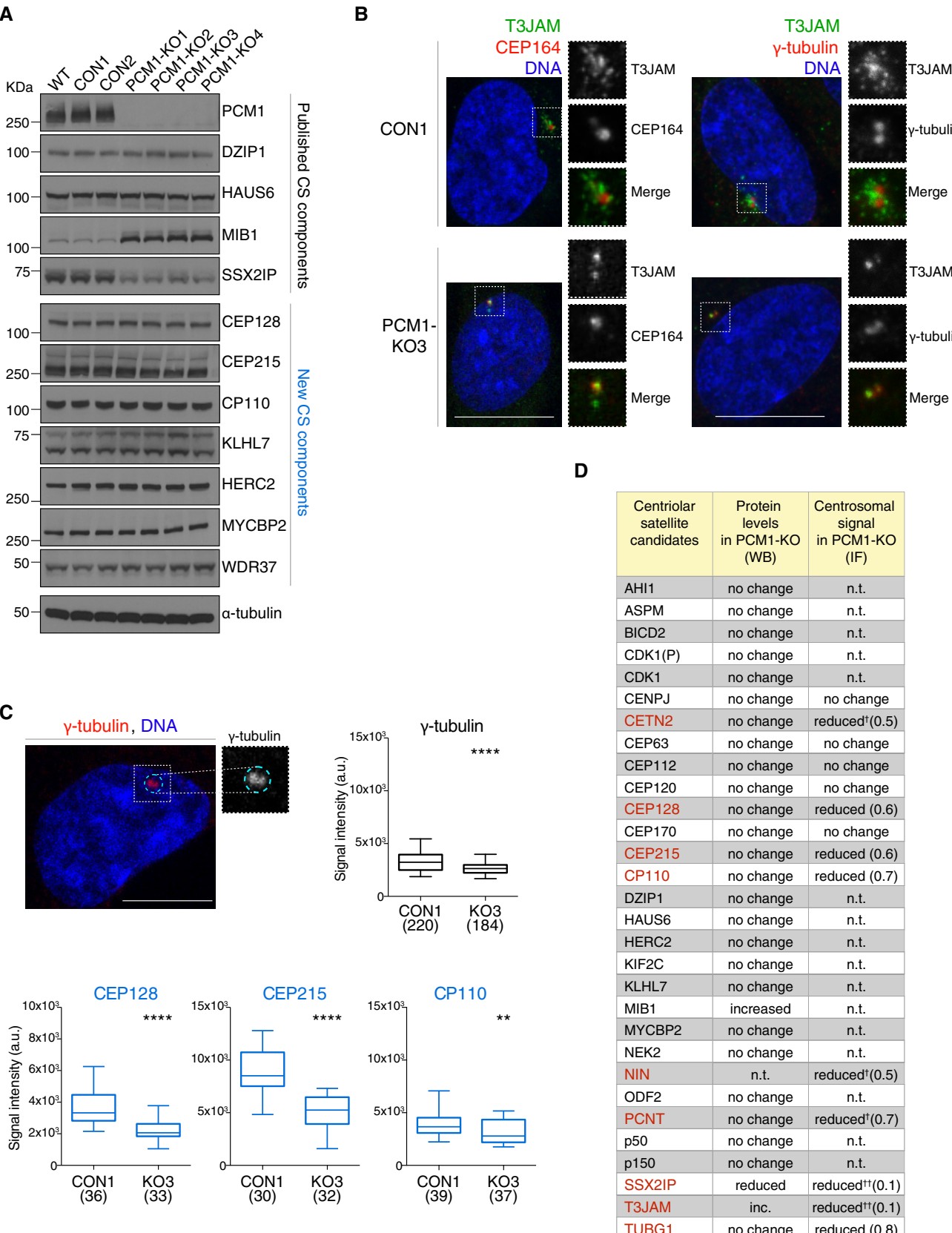

**Figure 7.**

**Figure 7. Differential effects of PCM1 loss on steady-state expression and centrosomal localisation of satellite components.**

A   Western blots show levels of known satellite components and satellite candidates (from CS-WT) in control (CON) and PCM1 knock-out (KO) RPE-1 cell clones. Western blots of further candidates are included in Appendix Fig S3. WT corresponds to parental RPE-1 cells. α-tubulin served as loading control.

B   Representative immunofluorescence images of T3JAM localisation in control (CON) and PCM1-KO (KO3) cells. Higher magnifications of framed areas are shown in right panels. Cells were co-stained with antibodies against T3JAM (green) and CEP164 (red), or T3JAM (green) and γ-tubulin (red), as indicated. DNA is in blue. Images correspond to the maximum intensity projection of the confocal micrograph. Scale bar: 10 μm.

C   Quantification of centrosomal enrichment of satellite candidates in PCM1-KO cells. As illustrated by the dotted line shown in cyan in the representative confocal image, signal intensities were measured on maximum intensity projection images within a circle encompassing the centrosome as defined by γ-tubulin-positive staining (γ-tubulin is shown in red, DNA in blue). Box plots show total centrosomal signal intensities of satellite candidates (CEP128, CEP215 and CP110) in control (CON1) and PCM1-KO (KO3) cells. Boxes represent interquartile ranges, lines in boxes the medians, and whiskers indicate the $5^{th}$–$95^{th}$ percentiles. P values were obtained by the Mann–Whitney $U$ test (**$P \leq 0.01$; ****$P \leq 0.0001$; n.s.: not significant); the number of cells analysed is reported in parentheses.

D   Table summarises protein levels and centrosomal signals of satellite candidates in PCM1-KO cells (primary data in (A), (B) and Appendix Fig S3). Proteins with change in centrosomal signal (P value of 0.01 or less, Mann–Whitney test) are shown in red. Numbers in parentheses depict fold changes in centrosomal signal intensities. [†]Satellite components centrin 3 (CETN3), ninein (NIN) and pericentrin (PCNT) were included as positive controls (Dammermann & Merdes, 2002). [††]For proteins with prominent satellite localisation (SSX2IP and T3JAM), reduction in centrosomal signal in PCM1-KO may correspond to loss of pericentrosomal satellites.

normal expression in acentriolar cells, CEP41, CEP112 and CEP135 were markedly reduced in SILAC-CS-STIL, indicative of their stable, centriole-dependent association with PCM1.

How centriolar satellites assemble is not well understood, but PCM1 is undoubtedly a major factor (Hori & Toda, 2017). Although PCM1 levels are constant during the cell cycle, satellite dispersal is triggered in mitosis by the dual specificity kinase DYRK3 (Rai et al, 2018). Because DYRK3 acts as a dissolvase for multiple membrane-less organelles, these recent findings raise the intriguing possibility that centriolar satellite formation may involve liquid–liquid phase separation (Zwicker et al, 2014), a process that could be driven by PCM1 oligomerisation (Dammermann & Merdes, 2002; Kubo & Tsukita, 2003). DYRK3 is not part of the CS-WT proteome, but since its association with satellites is likely to induce rapid dissolution of the organelle, it may not be possible to isolate DYRK3-bound granules.

A remarkable 82 of the 223 proteins in CS-WT are known components of the centrosome. Although several factors involved in centriole biogenesis are among these 82 proteins, the three critical proteins that drive centriole formation, SAS-6, STIL and CEP152 along with PLK4 were absent from CS-WT (Banterle & Gonczy, 2017; Nigg & Holland, 2018). The low detection rates may simply reflect low abundance of these proteins, which is especially relevant because STIL and PLK4 are among the five least abundant proteins at the centrosome according to a recent study (Bauer et al, 2016). Ranking proteins based on their estimated centrosomal copy numbers revealed that the presence of centrosomal proteins in our filtered dataset and CS-WT (nomenclature as in Fig 2A; Fig EV6 and Table EV4) is associated with higher centrosomal abundance. However, the second most abundant protein CEP152 does not survive filtering, whereas well-established satellite components such as CEP290 or KIAA0586/TALPID3 are present in CS-WT despite ranking in the bottom 10% for centrosomal abundancy. These exceptions imply that centrosomal abundance alone cannot predict whether a protein is detected in centriolar satellites, and thus, additional regulatory mechanisms must exist to account for satellite enrichment of some centrosomal proteins. Although the master kinase responsible for initiation of centriole assembly, PLK4, was detected only in CS-STIL (Bettencourt-Dias et al, 2005; Habedanck et al, 2005), its association with centriolar satellites is likely to be genuine; PLK4 has been shown to interact with and phosphorylate PCM1 and hence stabilise satellites (Firat-Karalar et al, 2014; Hori et al, 2016). The overall increase in PLK4 levels in STIL-KO[PCM1-GFP]

cells must have facilitated detection of this low-abundance kinase in CS-STIL (Arquint et al, 2015; Moyer et al, 2015). PCM1 aggregates that form in SSX2IP-depleted cells have been shown to contain several centrosomal proteins (i.e. centrobin, centrin, C-NAP1) with the notable exceptions of PLK4, CEP152 and SAS-6 (Hori et al, 2015). This lends support to our findings that the core centriole initiation network, and especially CEP152, is present at very low levels on centriolar satellites. Given the massive overlap between the centrosome and satellite proteomes, exclusion of the centriole assembly-initiating module from centriolar satellites could play an important role in preventing de novo formation of centrioles at these sites. De-regulation of this core module in tumours could drive centrosome amplification in a satellite-dependent manner. Intriguingly, an increase in PCM1-positive granules precedes ionising radiation-induced centrosome amplification (Loffler et al, 2013).

Centriolar satellites promote and suppress ciliogenesis in a context-dependent manner (Hori & Toda, 2017). Indeed, CS-WT contains many essential ciliogenesis factors, despite the proteome being derived from chicken DT40 B lymphocytes. Although normal lymphocytes do not bear primary cilia, similar to other transformed lines, serum starvation induces cilia formation in a small percentage of DT40 cells (Prosser & Morrison, 2015; de la Roche et al, 2016). In CS-WT, we detected five members of the octameric BBSome complex (2/4/7/8/9), including BBS4, a previously described satellite component (Stowe et al, 2012; Chamling et al, 2014). In addition to the BBSome, CS-WT contained several disease-linked regulators of ciliogenesis including IFT74, KLHL7, CEP41 and CENPF, which had no previous links to centriolar satellites, or with the exception of CEP41, to centrosomes. Functional cilia are crucial for Hedgehog signalling; indeed, a recent genome-wide screen for Hedgehog regulators identified many ciliogenesis-related factors (Breslow et al, 2018). Remarkably, 30 of these overlapped with the CS-WT proteome, including TEDC1, a new regulator of centriole assembly and ciliogenesis (Fig EV2E). It will be important to establish whether these ciliogenesis factors associate with centriolar satellites both in untransformed ciliating and non-ciliating cells, and whether their satellite localisation is relevant to cilia-related or cilia-independent functions (Novas et al, 2015; de la Roche et al, 2016).

Depletion or deletion of PCM1 in RPE-1 cells diminishes cilia numbers, but co-depletion of the MIB1 E3 ubiquitin ligase with PCM1 has been shown to partially restore ciliation (Wang et al, 2016). MIB1 suppresses ciliogenesis by destabilising TALPID3, a centriole distal end protein implicated in ciliary vesicle recruitment

(Kobayashi *et al*, 2014). Thus, a key role of satellites, at least in RPE-1 cells, is to protect centrosomal TALPID3 via sequestration of MIB1. Intriguingly, in CS-WT we detect both MIB1 and its target, TALPID3, and therefore, MIB1 is unlikely to effectively target TALPID3 for degradation at this location. Perhaps, they associate with distinct satellite granules; this could be brought about by active sorting or evolve by MIB1-driven degradation of TALPID3 on granules they initially co-habit. It is however also feasible that MIB1 activity is suppressed on centriolar satellites; its turnover is undoubtedly controlled by satellites and/or PCM1, because MIB1 levels are elevated in PCM1-KO cells. Compelling recent studies also implicate PCM1 in autophagy (Tang *et al*, 2013; Joachim & Tooze, 2016; Joachim *et al*, 2017). Indeed, we have identified several satellite components with links to autophagy including the essential autophagy regulator Hippo kinase STK4, its activator SAV1 and T3JAM (Peng *et al*, 2015; Wilkinson *et al*, 2015). The presence of E3 ubiquitin ligases in CS-WT further strengthens the notion that centriolar satellites may compartmentalise and control protein degradation pathways and hence contribute to developmental and pathogenic centrosome- and cilia-related processes (Lecland & Merdes, 2018).

# Materials and Methods

### Cell culture

DT40 and Jurkat/Clone E6-1 cells were grown in suspension as in Sir *et al* (2013) and Zyss *et al* (2011). HEK 293T/17 (HEK 293T) cells were grown in DMEM GlutaMAX (Gibco) supplemented with 10% FBS or tetracycline-free FBS (PAN Biotech UK Ltd) for Flp-In T-REx (Invitrogen) cells. hTERT RPE-1 (RPE-1) cell lines were grown as described in Joseph *et al* (2018).

For stable isotope labelling with amino acids in culture (SILAC) experiments, DT40 wild-type (WT) and STIL knock-out (KO) cell lines were cultured for 12 days in SILAC RPMI-1640 without lysine and arginine (Thermo Fisher Scientific); supplemented with either 800 μM light lysine ($^{14}N_2^{12}C_6$) and 482 μM light arginine ($^{14}N_4^{12}C_6$), or 800 μM heavy lysine ($^{15}N_2^{13}C_6$) and 482 μM heavy arginine ($^{15}N_4^{13}C_6$; Thermo Fisher Scientific), 10% dialysed FBS for SILAC (Thermo Fisher Scientific); and subjected to heat inactivation at 56°C for 20 min, 100 U/ml penicillin and 100 μg/ml streptomycin (Gibco). The incorporation of labelled amino acids was verified by mass spectrometry.

### Antibodies and immunostainings

DT40 and Jurkat cells were settled onto poly-L-lysine (Sigma-Aldrich)-coated glass coverslips, whereas HEK 293T, 293 Flp-In T-REx and RPE-1 cell lines were grown on uncoated coverslips. Cells were fixed with 100% −20°C methanol or with 4% paraformaldehyde (Polysciences) in phosphate-buffered saline (PBS) for 10 min followed by 5 min in 100% 20°C methanol (ACROS Organics). Cells were permeabilised in PBS-0.5% Tween-20 (Promega; PBS-T) or, for centriolar staining, in 0.5% Triton X-100 (ACROS Organics), 0.05% sodium dodecyl sulphate (SDS; Sigma-Aldrich) and 0.5% Tween-20 (Promega) in PBS for 5 min. Cells were then stained as described in Sir *et al* (2013). Primary antibodies are listed in the

reagents and tools table in Appendix Supplementary Methods. Secondary antibodies conjugated to Alexa Fluor 488, 555 or 647 were obtained from Invitrogen. To visualise DNA, coverslips were incubated with 1 μg/ml Hoechst 33258 (Sigma-Aldrich) and then mounted on glass slides (SuperFrost Ultra Plus, Thermo Scientific) using the ProLong Diamond Antifade Mountant (Invitrogen).

### Image acquisition and processing

Confocal images of fixed cells were taken using the Confocal White Light Laser (WLL) Leica TCS SP8 Microscope. All the images were acquired as *z*-stacks (0.5 μm step size) and taken with the HC Plan Apo 100×/1.40 OIL (CS2) objective. Image acquisition was carried out with the Leica Application Suite X (LAS X) software (Leica Microsystems).

Wide-field images of fixed cells were acquired as *z*-stacks (0.3 μm step size) using a Nikon Eclipse TE2000 Inverted Microscope with Neo 5.5 sCMOS camera (Andor) and Plan Apo VC 60× or 100×/1.40 OIL objectives. Following acquisition, images were imported into Fiji (2.0.0-rc-59/1.51k) or Volocity 6.0 (Perkin Elmer) to obtain maximum intensity projections of *z*-stacks. Images were then imported into Photoshop (Adobe CC 2017) and adjusted to use full range of pixel intensities. Images from each biological replicate were acquired using the same settings and processed in the same manner. For image analysis, see Appendix Supplementary Methods.

### Centriolar satellite isolation

Centriolar satellites were isolated based on the protocol described by Kim *et al* (2008). Briefly, $1.5 \times 10^9$ DT40 cells were treated with 2 μg/ml nocodazole and 1 μg/ml cytochalasin-B for 2 h, washed in PBS and lysed in centriolar satellite buffer [50 mM Tris–HCl pH 8.0, 150 mM NaCl, 1 mM EGTA, 1 mM MgCl$_2$, 10% glycerol, 0.1% NP-40, 1 mM dithiothreitol (DTT), protease inhibitor cocktail (Complete EDTA-free, Roche Diagnostics) and phosphatase inhibitor cocktail (PhosStop, Roche Diagnostics)] with a homogeniser. For SILAC experiments, an equal number ($0.75 \times 10^9$) of heavy (or light)-labelled DT40 WT$^{PCM1-GFP}$ and light (or heavy)-labelled DT40 STIL-KO$^{PCM1-GFP}$ cells were mixed before the lysis but following the nocodazole/cytochalasin-B treatment.

Lysates were cleared by centrifugation at 1,500 *g* for 5 min at 4°C. DNase I (Sigma-Aldrich) was added to the supernatants at 2 μg/ml for 15 min. Supernatants were then filtered through a 70-μm cell strainer (BD Falcon) and fractionated on discontinuous sucrose gradients at 100,000 *g* for 16 h at 4°C in the SW40 Ti rotor (Beckman Coulter). The discontinuous sucrose gradients were prepared layering 1 ml of each 10–70% (w/v) sucrose solution in Ultra-Clear Tubes (14 × 95 mm, Beckman). The sucrose solutions were prepared in 50 mM Tris–HCl pH 8.0, 150 mM NaCl, 1 mM EGTA, 1 mM MgCl$_2$, 10% glycerol, 0.05% NP-40, 1 mM DTT, protease and phosphatase inhibitor cocktails. After the sucrose density gradient centrifugation, each 1 ml sucrose fraction was collected from the top of the gradient and transferred in a separate 1.5-ml tube.

For immunoprecipitations, GFP, PCM1 antibodies or rabbit IgGs (60 μg) were coupled to 200 μl of magnetic protein G Dynabeads (Invitrogen). Following overnight incubation, antibodies were cross-linked to the beads in 40 mM DMP-100 mM Na-tetraborate. Reaction was stopped by adding 1 M Tris–HCl (pH 8.0) and washed

three times PBS-0.01% Tween-20 (Promega). 30, 40 and 50% sucrose fractions were pooled for immunoprecipitations using magnetic protein G Dynabeads (Invitrogen). CS-enriched fractions (30, 40, 50%) were pooled, diluted in centriolar satellite buffer (to have maximum 10% (w/v) as final concentration of sucrose) and incubated with beads for 3 h at 4°C. The beads were then washed five times at 4°C with 50 mM Tris–HCl pH 8.0, 150 mM NaCl, 1 mM EGTA, 1 mM $MgCl_2$, 5% glycerol, 0.1% NP-40, 1 mM DTT, protease and phosphatase inhibitors (Roche Diagnostics) and then resuspended in NuPAGE LDS Sample Buffer (Invitrogen), supplemented with NuPAGE Sample Reducing Agent (Invitrogen) and vortexed multiple times. The supernatant was heated to 80°C for 10 min. For Western blotting, sucrose fractions were subjected to chloroform/methanol precipitation. Protein pellets were then resuspended in NuPAGE LDS Sample Buffer (Invitrogen), supplemented with NuPAGE Sample Reducing Agent.

### Liquid chromatography-tandem mass spectrometry (LC-MS/MS) and analysis

Sample preparation is described in Appendix Supplementary Methods. LC-MS analysis was performed using the Dionex Ultimate 3000 UHPLC system coupled with the Orbitrap Velos mass spectrometer (Thermo Scientific). Mass spectra were acquired in data-dependent mode using a "top20" method. Whole-cell proteome samples were measured by a Q Exactive or Q Exactive HF Orbitrap mass spectrometer. MaxQuant software data processing and bioinformatic data analysis is described in detail in Appendix Supplementary Methods.

### RNA interference

RPE-1 cells were transfected with Lipofectamine RNAiMAX reagent (Thermo Fisher Scientific), following manufacturer's instructions. siRNAs were used at a final concentration of 60 nM, and the siRNA treatments were carried out for 72 h after transfection. siRNA sequences are listed in the Reagents and tools table in Appendix Supplementary Methods.

### Statistical analysis

Statistical analysis for Figs 4C, 5A and C, and 7C, and Appendix Fig S3 was performed with GraphPrism 6.0 (GraphPad). Details of statistical tests are highlighted in relevant figure legends. Statistical analyses for mass spectrometry data are detailed in Appendix Supplementary Methods.

## Data availability

The mass spectrometry data from this publication have been deposited in the ProteomeXchange Consortium (http://proteomecentral.proteomexchange.org) via the PRIDE partner repository with the dataset identifiers PXD010325 (CS proteome), PXD011248 (WCP SILAC) and PXD011247 (SILAC-CS). See Appendix for further details.

**Expanded View** for this article is available online.

## Acknowledgements

The authors would like to thank Lovorka Stojic for the critical reading of the manuscript, and Shankar Balasubramanian and Thomas Mayer for reagents. We also thank members of the Microscopy and Proteomics Core facilities of CRUK CI, and in particular Valar Nila Roamio Franklin for processing samples. JVK is an emeritus staff member of the MRC Laboratory of Molecular Biology and is very grateful to Dr Sean Munro for providing a bench and support. This work was funded by Cancer Research UK (C14303/A17197 to M.M and C14303/A17043 to F.G). J.T was supported by a Deutsche Forschungsgemeinschaft research fellowship, HA-8069/1-1. We acknowledge support from NIHR Cambridge Biomedical Research Centre, the University of Cambridge and Hutchison Whampoa Ltd.

## Author contributions

Experiments were designed by FG and VQ and performed by VQ, JT and JVK. mass spectrometry was conducted by CGT and EKP with help from CSD. Quantitative mass spectrometry analysis was performed by J-XC with advice by MLM. All authors contributed to writing and editing the article.

## Conflict of interest

The authors declare that they have no conflict of interest.

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
