## [Review Process File · The EMBO Journal]

Centriolar satellites are acentriolar assemblies of centrosomal proteins

Valentina Quarantotti, Jia-Xuan Chen, Julia Tischer, Carmen Gonzalez Tejado, Evaggelia K. Papachristou, Clive S. D'Santos, John V. Kilmartin, Martin L. Miller and Fanni Gergely.

Review timeline:

Submission date:	13 th November 2018
Editorial Decision:	13 th December 2018
Revision received:	15 th March 2019
Editorial Decision:	10 th April 2019
Revision received:	24 th April 2019
Accepted:	6 th May 2019

Editor: Hartmut Vodermaier

Transaction Report:

1st Editorial Decision

13th December 2018

Thank you again for submitting your manuscript on centriolar satellite composition to by The EMBO Journal. We have now received the comments from two expert referees, copied below for your information, in whose light we shall be happy to consider a revised version further for publication as a Resource Article. As you will see, both referees do raise a number of specific issues, which I would like to invite you to address during such a revision.

Regarding nomenclature (which was in any case differing between your article and the cosubmitted manuscript), I agree with referee 2 that it would be preferable to use the term 'centriolar satellites' instead of pericentriolar satellites throughout, and to avoid unnecessary abbreviations such as PS or CS (at least in the manuscript text)

REFeree REPORTS

Referee #1:

Valentina Quarantotti et al, have developed an elegant experimental system to study pericentriolar satellites (PS) based on affinity enrichment of proteins interacting with endogenously GFP-tagged PCM1 in DT40 chicken B lymphocytes. Using sucrose gradient centrifugation prior to the affinity enrichment, they circumvent co-purification of centrosomes and by performing experiments in cells lacking centrioles by STIL or Cep152 depletion they are able to assess PS protein composition and assemble independently of centrosomes. The affinity enrichment experiments identify a PCM1 interactome consisting of 223 proteins enriched in previously reported PS, centrosome, and ciliary proteins. Several of the identified candidate proteins are tested in human cells for PCM1 co-localization, affinity enrichment evaluated by Western blotting, function in ciliogenesis, and subcellular redistribution after PCM1 depletion. This validates novel PS components and reveals candidates that positively regulate cilia assembly but are dispensable for PS integrity.

Importantly, 170 proteins of the significantly enriched proteins are shared between the WT, STIL-KO and CEP152-KO affinity enrichment experiments. This suggests that the majority of PS-proteins remains associated with PCM1 in cells lacking centrosomes. These observations are elaborated further by direct quantitative comparison of the whole cell proteome and PCM1-interactome in normal and STIL depleted cells. Although a subset of PS and centrosome proteins display changes in expression levels or association with PCM1, the majority of proteins are confirmed to remain stably associated with PS. They conclude that PS form independently of centrosomes and associate with a distinct pool of centrosome proteins that could provide a reservoir of proteins when transcription or translation of centrosome genes is restricted. Several centrosome proteins required for centriole biogenesis are observed to be excluded from the distinct pool of PS-associated proteins, and they put forward the interesting suggestion that this could provide a mechanism to prevent aberrant centriole assembly at the sites of PS.

Comments

1. Line 188: "In summary, we confirmed co-immunoprecipitation and co-localisation between several new PS candidates and PCM1 in multiple human cell lines". It would be helpful with a more specific conclusion on the proteins that are convincingly validated as novel PS components. Alternatively, a table could be included that summarises the proteins tested, the criteria used, and the status of validation.
2. In this study, the interactome of PS is compared for normal and acentriolar cells pre-treated with nocodazole and cytochalasin. Considering that PS are dispersed upon centriole loss or microtubule disassembly, it should be taken into account and discussed that the direct effect on PS integrity upon centriole loss is not resolved from the dispersion effect of microtubule disassembly.

Referee #2:

Centriolar satellites are cytoplasmic structures linked to a wide variety of cellular processes including the assembly and function of centrosomes as well as cilia, although their precise contribution to these processes remains unclear. A core structural component of satellites is PCM1, in the absence of which satellites are thought to be lost. Here, Quarantotti and colleagues seek to address the composition and function of satellites by performing sucrose sedimentation and PCM1 immunoprecipitation from wild-type and acentriolar cells. They further compare the protein complement of wild-type and PCM1-deficient cells. These experiments allow them to identify a number of novel centriolar satellite components as well as conclude that satellites and centrosomes form independently, but share a substantial fraction of their proteomes. Overall, these experiments are well executed and while there are no great surprises, the results obtained should be of sufficient interest to the readership of the EMBO Journal. Nevertheless, as detailed below a number of weaknesses remain that need to be addressed prior to publication.

Major points

1. The authors' approach, partial purification of satellites followed by PCM1 immunoprecipitation using GFP antibodies and tagging of the endogenous copy of PCM1, complements previous work by Gupta et al (Cell 2015) using FLAG-tag immunoprecipitation as well as PCM1 BioID. As shown in Fig EV4A, they identify a considerably larger set of potential interactors including numerous centrosomal proteins, which could reflect improved isolation of satellites or less stringent filtering of the data. Since these interactors remain in acentriolar cells, the authors conclude that centrosomes and satellites share much of their protein composition. Since the authors also demonstrate cell type-dependent differences in satellite localization (Figs 4, EV5), one possibility is that all centrosomal proteins to a greater or lesser extent localize to satellites. If so, their detection may just be a question of protein abundance. This has been measured for numerous centrosomal proteins by Bauer et al, EMBO J 2016. A comparison with their study may help to address whether the authors' failure to detect specific proteins such as the core centriole initiation network of PLK4, CEP152 and SAS-6 (p12) simply reflects their low copy number at centrosomes/in the cell.
2. In their study, the authors report the identification of a number of novel centriolar satellite components, many of which had previously been mapped to specific centrosomal

subcompartments/functions (Fig 3E), while others (eg T3JAM, Fig 4A) are apparently novel. The authors here significantly understate the number of previously known satellite components (estimated at >100 by Hori and Toda, *Cell Mol Life Sci* 2017). Thus, for example, centrin-3/CTN3, ninein and pericentrin/PCNT were already identified as satellite components in one of the earliest studies (Dammermann and Merdes, *JCB* 2002), which the authors themselves refer to in their introduction (p4). Similarly, CEP350, OFIP/KIAA0753 and CEP63 were identified as a satellite components in the review by Hori and Toda. A more stringent definition of novel components would be in the best interests of the authors by more clearly highlighting the truly novel components such as T3JAM, WDR37 and CCDC77.

3. While the authors validate their siRNA-mediated depletion of novel components by qPCR (Fig EV6), they do not validate the satellite signal they observe by immunofluorescence eg in Fig 4. This is an important point if they wish to convince the reader of the validity of their identifying these components as satellite proteins.

Other points

4. I am not a great fan of the authors referring to centriolar satellites as pericentriolar satellites or PSs. While neither term is particularly accurate, it would be good to stick to the generally accepted term to avoid confusion. Further, abbreviations like PS or CS are rarely helpful, especially when 'satellite' would hardly take up much more space.

5. On p5 the authors write "In sucrose density gradient centrifugation PCM1-GFP and the PS-associated ubiquitin ligase MIB1 were found in fractions 30-70%, whereas fractions 60-70% contained the centrosomal proteins centrin-2 and γ -tubulin (Fig 1F). Therefore, to minimise co-isolation of centrosomes, only fractions 30-50% were pooled for subsequent GFP and control IgG immunoprecipitations (Fig 1F)." It may indeed be true that centrosomes are present in fractions 60-70% based on other data. However, the presence of centrin-2 and γ -tubulin in those fractions cannot be indicative of the presence of centrosomes as these are absent in STIL and CEP152 knock-outs while the centrin/ γ -tubulin bands clearly persist.

6. The color merge for CEP215 in Fig EV5 appears to be incorrect.

7. In Fig 6 the authors draw attention to a decrease in centrosomal signal of γ -tubulin following PCM1 knockout. This decrease, while statistically significant, is hardly prominent compared to the lack of accumulation of other centrosomal components like CEP215 and may be an indirect consequence of the loss of such scaffolding components. The authors may wish to discuss this possibility. Further, it would perhaps be worthwhile to discuss the apparent absence of γ -tubulin from centriolar satellites, despite it being one of the most abundant proteins at centrosomes (Bauer et al, *EMBO J* 2016).

Response to the Referees' comment

(Quarantotti et al., Centriolar satellites areacentriolar assemblies of centrosomal proteins)

We are grateful to the Reviewers for their constructive critique. In response to the comments we have introduced changes to the text (major changes are highlighted in blue) and included new figures, data and tables (Figs. 4C, 4D, 5C, 5D, 7D, EV6, Appendix Fig S2). We hope that the Reviewers agree that the manuscript has improved as a direct result of their input.

We have listed our point by point response below.

Referee #1:

Valentina Quarantotti et al, have developed an elegant experimental system to study pericentriolar satellites (PS) based on affinity enrichment of proteins interacting with endogenously GFP-tagged PCMI in DT40 chicken B lymphocytes. Using sucrose gradient centrifugation prior to the affinity enrichment, they circumvent co-purification of centrosomes and by performing experiments in cells lacking centrioles by STIL or Cep152 depletion they are able to assess PS protein composition and assemble independently of centrosomes. The affinity enrichment experiments identify a PCMI interactome consisting of 223 proteins enriched in previously reported PS, centrosome, and ciliary proteins. Several of the identified candidate proteins are tested in human cells for PCMI co-localization, affinity enrichment evaluated by Western blotting, function in ciliogenesis, and subcellular redistribution after PCMI depletion. This validates novel PS components and reveals candidates that positively regulate cilia assembly but are dispensable for PS integrity.

Importantly, 170 proteins of the significantly enriched proteins are shared between the WT, STIL-KO and CEP152-KO affinity enrichment experiments. This suggests that the majority of PS-proteins remains associated with PCMI in cells lacking centrioles. These observations are elaborated further by direct quantitative comparison of the whole cell proteome and PCMI-interactome in normal and STIL depleted cells. Although a subset of PS and centrosome proteins display changes in expression levels or association with PCMI, the majority of proteins are confirmed to remain stably associated with PS. They conclude that PS form independently of centrosomes and associate with a distinct pool of centrosome proteins that could provide a reservoir of proteins when transcription or translation of centrosome genes is restricted. Several centrosome proteins required for centriole biogenesis are observed to be excluded from the distinct pool of PS-associated proteins, and they put forward the interesting suggestion that this could provide a mechanism to prevent aberrant centriole assembly at the sites of PS.

We are pleased that the Reviewer finds our experimental system elegant and our discussion interesting, and are grateful for the insightful comments.

Comments

1. Line 188: "In summary, we confirmed co-immunoprecipitation and co-localisation between several new PS candidates and PCMI in multiple human cell lines". It would be helpful with a more specific conclusion on the proteins that are convincingly validated as novel PS components. Alternatively, a table could be included that summarises the proteins tested, the criteria used, and the status of validation.

We thank the Reviewer for this useful suggestion. We have produced a table that contains the information requested; for each candidate investigated, we state the cell lines used, the method of validation and the outcome (Figure 5D). We have included an additional table (Figure 7D) that summarises results from the PCMI knockout cell lines shown previously in Figs 6 and EV9 (now Figs 7, EV5 and Appendix Fig S3). We also take care throughout the revised manuscript to refer to proteins found in our proteomic survey as centriolar satellite **candidates** unless validated by western blotting/immunoprecipitation and/or immunofluorescence.

2. In this study, the interactome of PS is compared for normal and acentriolar cells pre-treated with nocodazole and cytochalasin. Considering that PS are dispersed upon centriole loss or microtubule disassembly, it should be taken into account and discussed that the direct effect on PS integrity upon centriole loss is not resolved from the dispersion effect of microtubule disassembly.

We agree that this is an important point, and we considered this issue in our experimental design.

Briefly, under our experimental conditions centriolar satellite distribution appears different between acentriolar cells and nocodazole/cytochalasin-treated wild-type (WT) cells (representative images are now shown in Fig. EV1B, previously in Fig. EV2; compare DMSO-treated acentriolar and drug-treated WT cells). Whereas satellite dispersal is seen in 50% of acentriolar cells, nocodazole/cytochalasin treatment induces minor changes in satellite distribution in WT cells; large granules persist or continue to form, and a relatively minor reduction in small granules is observed. By contrast, nocodazole/cytochalasin increases the size of satellite granules in acentriolar cells, thereby achieving a more comparable satellite size distribution between acentriolar and WT cells. Lack of satellite dispersal may be due to incomplete depolymerisation of microtubules by nocodazole; in our hands complete elimination of interphase microtubules in DT40 requires prolonged incubation of cells on ice after nocodazole treatment.

We obviously cannot exclude that nocodazole/cytochalasin treatment influences satellite association of certain proteins, and if it does, its effects may impact centriolar satellites differently in WT and acentriolar cells. Nonetheless, our SILAC data in Fig. 6E demonstrates that with GFP antibodies we could capture equal amounts of PCM1-GFP from the 30-50% sucrose fractions of WT and acentriolar STIL cells. Moreover, quantitative analysis revealed no change in the majority of PCM1-associated factors, although there were some notable differences. Reassuringly, the latter reflected changes also seen in the whole cell proteome (obtained from untreated cells) of the corresponding genotype, suggesting that in these cases drug treatment did not mask alterations (i.e. PLK4 increase in STIL-KO WCP and satellites).

We have now included an additional sentence to clarify this point:

Line 95: This drug combination increased centriolar satellite clustering in acentriolar cells without causing satellite dispersal in WT.

We have also added the following description to the legend of Figure EV1B:

Asterisks mark cells with dispersed satellites. Note that drug treatment leads to an increase in large and a decrease in small satellite granules in all three genotypes, but the effects are more prominent in acentriolar than in WT cells.

Referee #2:

Centriolar satellites are cytoplasmic structures linked to a wide variety of cellular processes including the assembly and function of centrosomes as well as cilia, although their precise contribution to these processes remains unclear. A core structural component of satellites is PCMI, in the absence of which satellites are thought to be lost. Here, Quarantotti and colleagues seek to address the composition and function of satellites by performing sucrose sedimentation and PCMI immunoprecipitation from wild-type and acentriolar cells. They further compare the protein complement of wild-type and PCMI-deficient cells. These experiments allow them to identify a number of novel centriolar satellite components as well as conclude that satellites and centrosomes form independently, but share a substantial fraction of their proteomes. Overall, these experiments are well executed and while there are no great surprises, the results obtained should be of sufficient interest to the readership of the EMBO Journal. Nevertheless, as detailed below a number of weaknesses remain that need to be addressed prior to publication.

We are pleased that the Reviewer finds our study well executed and interesting.

Major points

1. The authors' approach, partial purification of satellites followed by PCMI immunoprecipitation using GFP antibodies and tagging of the endogenous copy of PCMI, complements previous work by Gupta et al (Cell 2015) using FLAG-tag immunoprecipitation as well as PCMI BioID. As shown in Fig EV4A, they identify a considerably larger set of potential interactors including numerous centrosomal proteins, which could reflect improved isolation of satellites or less stringent filtering of the data. Since these interactors remain in acentriolar cells, the authors conclude that centrosomes and satellites share much of their protein composition. Since the authors also demonstrate cell type-dependent differences in satellite localization (Figs 4, EV5), one possibility is that all centrosomal proteins to a greater or lesser extent localize to satellites. If so, their detection may just be a question of protein abundance. This has been measured for numerous centrosomal proteins by Bauer et al, EMBO J 2016. A comparison with their study may help to address whether the authors' failure to detect specific proteins such as the core centriole initiation network of PLK4, CEP152 and SAS-6 (p12) simply reflects their low copy number at centrosomes/in the cell.

We are grateful to the Reviewer for this helpful and insightful comment.

We subjected our data to stringent filtering, which for clarity is now depicted in a flow chart (Fig. 2A). In addition, as suggested by the Reviewer, we evaluated presence vs absence of centrosomal proteins in our wild-type centriolar satellite datasets according to their centrosomal abundance as measured/estimated in the study by the Nigg lab (Bauer et al., 2016)(new Fig. EV6). A statistically significant positive association was seen between abundance of centrosomal proteins and their presence in the unfiltered, filtered and CS-WT datasets (terminology of datasets is depicted in Fig 2A). The strongest association was found for the filtered dataset, indicating that abundance is particularly relevant to detection of proteins across multiple biological replicates. A notable exception is CEP152, which is the second most abundant protein in the centrosome according to the Bauer study, yet, is detected only in the unfiltered data. Consistent with their very low abundance, no peptides were found for STIL and PLK4 in our datasets, but on the other hand, CEP290 and TALPID3 are present in CS-WT despite being very scarce. We have updated the discussion according to these findings and included a new figure with these comparisons and a table (line 376, Fig. EV6, Table EV4).

2. In their study, the authors report the identification of a number of novel centriolar satellite components, many of which had previously been mapped to specific centrosomal subcompartments/functions (Fig 3E), while others (e.g. T3JAM, Fig 4A) are apparently novel. The authors here significantly understate the number of previously known satellite components (estimated at >100 by Hori and Toda, Cell Mol Life Sci 2017). Thus, for example, centrin-3/CTN3, ninein and pericentrin/PCNT were already identified as satellite components in one of the earliest studies (Dammermann and Merdes, JCB 2002), which the authors themselves refer to in their introduction (p4). Similarly, CEP350, OFIP/KIAA0753 and CEP63 were identified as a satellite component in the review by Hori and Toda. A more stringent definition of novel components would

be in the best interests of the authors by more clearly highlighting the truly novel components such as T3JAM, WDR37 and CCDC77.

We apologise for this oversight and amended the table in Fig. 3B (entitled Published CS components) to include ninein, centrin-3 and pericentrin, as these were indeed shown to exhibit overlap with PCM1 by Dammermann and Merdes.

The table in Fig. 3B includes all the centriolar satellite components classified as components by Hori and Toda (Cell Mol Life Sci 2017, Fig. 1) with the exceptions of CEP350 and CCDC138. CEP350 was omitted because it was not shown to be a satellite component *per se* (As stated by the authors in their review: “...Other proteins including CAP350, FOP, Ninein, ODF2/Cenexin1 and Trichoplein, among which FOP is a centriolar satellite component, are also involved in microtubule anchoring...”). As for CCDC138, this protein was among the 100 or more proximity interactors of PCM1 identified by BioID (Gupta et al., Cell, 2015), but was not subjected to subsequent validation, unlike the other 8 proteins from the same study that are shown in our table. The table also includes proteins that have been reported to co-localise with PCM1 (i.e. CEP89, DISC1, CSPP1) but were not covered by Hori and Toda and factors that have been characterised as centriolar satellite components since the publication date of the review (i.e. DZIP1, ODF2L, VPS4). We therefore believe that the revised (to include ninein, centrin and pericentrin) table provides a reasonably accurate overview of centriolar satellite components.

Regarding the estimate of >100 satellite components, Hori and Toda based their statement on the aforementioned BioID study (Gupta et al., 2015). While Gupta et al indeed identified over 100 proteins using PCM1-BioID (we show comparisons between that dataset and ours in Fig. EV2), whether all these factors correspond to satellite components remains to be addressed. In particular, PCM1 has satellite and centrosomal pools, and therefore proximity labelling does not distinguish between PCM1 interactors at centrosomes or centriolar satellites (or additional locations). In addition, PCM1-BioID was performed by prolonged (24 hr) exogenous expression of the PCM1-BiR fusion product, which could also influence the interactome.

For sake of consistency, we refer to all the PCM1 interactors identified in our study as satellite candidates unless they were validated (see new table in Fig. 5D that summarises the various validation experiments). Given that not all known satellite components can be detected in satellites by immunofluorescence (e.g. CEP63, Firat Karalar et al., 2014), we also consider the western blots of PCM1 pulldowns performed from 30-50% sucrose fractions (from DT40 and HEK293) as evidence for PCM1 binding and satellite association.

3. While the authors validate their siRNA-mediated depletion of novel components by qPCR (Fig EV6), they do not validate the satellite signal they observe by immunofluorescence e.g. in Fig 4. This is an important point if they wish to convince the reader of the validity of their identifying these components as satellite proteins.

Prompted by the Reviewer’s comment, we performed immunofluorescence on RPE1 cells depleted of T3JAM and WDR37. We validated the staining pattern using siRNA-mediated depletion of T3JAM; we show representative immunofluorescence images as well as quantification of the signal in Fig. 4C. Note that the T3JAM antibody does not recognise the endogenous product in westerns.

For WDR37 when we stained depleted cells with the antibody used for the immunofluorescence data throughout our previous submission, we noted an overall reduction in WDR37 levels (Response Figure 1A, B). However, despite a depletion efficiency of 85%, a prominent centrosomal signal remained visible. Unfortunately, upon reordering this antibody for further replicates of the siRNA experiments (source: Atlas Prestige/Human Protein Atlas), it gave a diffuse membrane-like signal while still staining the centrosome, and these signals were resistant to siRNA (Response Figure 1C). Furthermore, the antibody that is specific in westerns shows a Golgi signal in immunofluorescence, which does not diminish upon siRNA. In Human Protein Atlas, a third WDR37 antibody exhibits a granular/vesicular staining that does not phenocopy satellites. A possible explanation for these discrepancies is that WDR37 has 7 different splice forms that may be present in different subcellular pools. However, given that the commercially available antibodies show inconsistent localisation patterns and a potentially non-specific centrosomal signal (WDR37 has not been found in the centrosome or cilia in published proteomic studies), we have decided to remove the WDR37

immunofluorescence data from the manuscript as they would not be reproducible with currently available reagents. These results are included for inspection by the Reviewer in Response Figure 1. Instead, we now show Flp-In T-REx HEK293 cells that inducibly express GFP fusions of satellite candidates TRIM37 and CEP170 (in addition to CCDC77) (Fig. 4D). Similar to other satellite components, these form cytoplasmic structures when overexpressed that partially or fully overlap with PCM1 signal.

Finally, to confirm that the PCM1 staining pattern in Jurkat cells indeed corresponds to centriolar satellites, cells were treated with nocodazole to depolymerise microtubules. Representative images of centrosomal regions from nocodazole-treated cells show loss of PCM1 signal from this area (Fig. EV3B).

Other points

4. I am not a great fan of the authors referring to centriolar satellites as pericentriolar satellites or PSs. While neither term is particularly accurate, it would be good to stick to the generally accepted term to avoid confusion. Further, abbreviations like PS or CS are rarely helpful, especially when 'satellite' would hardly take up much more space.

We have adapted the term centriolar satellites throughout the manuscript.

5. On p5 the authors write "In sucrose density gradient centrifugation PCM1-GFP and the PS-associated ubiquitin ligase MIB1 were found in fractions 30-70%, whereas fractions 60-70% contained the centrosomal proteins centrin-2 and γ -tubulin (Fig 1F). Therefore, to minimise co-isolation of centrosomes, only fractions 30-50% were pooled for subsequent GFP and control IgG immunoprecipitations (Fig 1F)." It may indeed be true that centrosomes are present in fractions 60-70% based on other data. However, the presence of centrin-2 and γ -tubulin in those fractions cannot be indicative of the presence of centrosomes as these are absent in STIL and CEP152 knock-outs while the centrin/ γ -tubulin bands clearly persist.

The Reviewer raises a valid point; it is counterintuitive that the 60-70% fractions, where one would expect to see the centrosome, contain two centrosomal markers (centrin and gamma tubulin) in acentriolar cells. These proteins may come from 1) very large centriolar satellites and/or 2) acentriolar PCM assemblies. We previously demonstrated that ~50% of acentriolar cells contain a single PCM focus, which contains varying amounts of PCM proteins (for CDK5RAP2/CEP215, pericentrin and gamma tubulin, see Sir et al., 2013; for centrin see Response Figure 2 and for gamma tubulin see Fig. 1E-G). Centrin and gamma tubulin in the high sucrose fractions of acentriolar cells may therefore originate from PCM assemblies, but given the association of these two proteins with centriolar satellites, we cannot exclude that larger satellites also sediment in these fractions (indeed, PCM1 is also present there). In Fig. 1H centrin appears reduced in the 60-70% fractions of acentriolar cells, which may represent loss of the centriole-associated pool.

To improve clarity, we have modified the text to:

Line 98: Centrosomes are expected to sediment in the higher sucrose fractions (Chavali & Gergely, 2015); indeed, the 60-70% fractions of WT cells appeared more enriched for the centriolar protein centrin 2 than the equivalent fractions in the acentriolar cells.

6. The color merge for CEP215 in Fig EV5 appears to be incorrect.

We have corrected this mistake.

7. In Fig 6 the authors draw attention to a decrease in centrosomal signal of γ -tubulin following PCM1 knockout. This decrease, while statistically significant, is hardly prominent compared to the lack of accumulation of other centrosomal components like CEP215 and may be an indirect consequence of the loss of such scaffolding components. The authors may wish to discuss this possibility. Further, it would perhaps be worthwhile to discuss the apparent absence of γ -tubulin from centriolar satellites, despite it being one of the most abundant proteins at centrosomes (Bauer et al, EMBO J 2016).

We now mention that reduction in pericentrin could impact on accumulation of gamma-tubulin; based on a recent paper from the Rios group CDK5RAP2 does not affect centrosomal gamma tubulin levels in RPE1 cells. Importantly, gamma tubulin is present in the CS-WT and CS-CEP152 proteomes (TUBG1); we have now included it in our schematic in Fig. 3E. We failed to see TUBG1 in CS-STIL, and this may be due to the 2-fold reduction of TUBG1 in CS-STIL according to the SILAC results.

Response Figure 1

Figures for referees not shown.

Response Figure 2

Figures for referees not shown.

Thank you for submitting your revised manuscript to The EMBO Journal. It has now been re-reviewed by one of the original referees, in light of whose positive comments we are now happy to offer publication of the study in our Resource section. As you will see, referee 1 retains some minor reservations and recommends addition of a cautionary note regarding the interpretation of the proteomics data, which I would like to invite you to incorporate during a final round of minor revision.

REFEREE REPORTS

Referee #1:

In the revised manuscript, it is most helpful that the authors have included a table that summarizes information for each of the centriolar satellite candidates investigated and to what extent the candidates have been validated as novel centriolar satellite proteins. Implementation of additional changes has contributed to the overall improvement of the revised manuscript.

Comments regarding the dispersal of centriolar satellites in the absence or presence of centrioles or in the absence or presence of actin filaments and microtubules is also well addressed by fluorescence microscopy images from the view point of PCM1 and its subcellular distribution. The presence of centriolar satellites in acentriolar cells indeed support the key conclusion that satellites assemble independently of centrosomes.

When the experimental conditions are extended to MS-based proteomics experiments to further study how centriole loss impact centriolar satellite assembly, it calls for a few additional comments. In these comparative affinity enrichment experiments, observed differences in satellite composition can indeed be ascribed to centriole loss because WT, STIL KO and Cep152 KO cells are all treated with nocodazole and cytochalasin. Absence of differences might, however, represent cases where the potential effect of centriole loss is masked by the effect of actin filament and microtubule depolymerisation. Moreover, in SILAC-based protein-protein interaction studies, stable vs dynamic association can be determined by comparing ratios measured for experiments where SILAC-labelled proteins are mixed before or after affinity enrichment. When mixed before affinity enrichment, dynamic interactors are likely to exchange, balancing the SILAC ratios towards those observed between the whole cell proteomes. In this study, SILAC-labelled proteins are mixed for an extended period of time and most proteomes have ratios similar to those observed for the whole cell proteomes. Again, this calls for a cautionary note on the interpretation and conclusion that the centriolar satellite proteome is largely unaltered in acentriolar cells. At the same time, it emphasizes the significance of Cep41 observed with a low STIL-KO/WT SILAC ratio, which exemplify the interplay between centrosomes and centriolar satellites. For Cep41, the authors point out the consistency between the label-free (Fig. 2C) and the SILAC-based experiments (Fig. 6E). Provided the SILAC-labelled protein exchange issue mentioned above, it would be of interest to indicate if this is also the case for the other satellite proteins.

We are pleased about the positive comments and your offer of accepting our paper subject to the minor corrections.

We have also considered the Reviewer's comments in our revision and agree with both points the Reviewer has made. We have therefore inserted the recommended cautionary notes into the second paragraph of our Discussion, which now reads as:

Quantitative SILAC analysis uncovered relatively few differences between satellite composition of WT and STIL-KO cells; however, two aspects of the experimental design could have caused compression of SILAC ratios. First, satellite isolation was performed from cells

with partially depolymerised microtubule and actin networks, which may have masked some of the differences between control and acentriolar satellite proteomes. Second, the mixing of light and heavy labelled lysates prior to satellite purification potentially enabled exchange between differentially labelled forms of satellite components, thus compressing SILAC ratios of dynamic interactors of PCMI (Mousson et al., 2008). Nonetheless, for the subset of proteins we could quantitate both in whole cell proteomes and centriolar satellites of WT and STIL-KO cells, the changes in total levels did not exceed those in satellite pools. Interestingly, despite normal expression in acentriolar cells, CEP41, CEP112 and CEP135 were markedly reduced in SILAC-CS-STIL, indicative of their stable, centriole-dependent association with PCMI.

In addition, for improved clarity we inserted this line into the legend of Fig. 6A:

Note that differentially labelled cells were mixed prior to the multistep CS purification process to minimise variability in sample preparation.

Accepted

6th May 2019

Thank you for submitting your final revised manuscript for our consideration. I am pleased to inform you that we have now accepted it for publication in The EMBO Journal.

Corresponding Author Name: Fanni Gergely

Journal Submitted to: The EMBO Journal

Manuscript Number: EMBOJ-2018-101082R